🍧 PLOS | ONE

# Identifying maintenance hosts for infection with *Dichelobacter nodosus* in free-ranging wild ruminants in Switzerland: A prevalence study

Gaia Moore-Jones[1], Flurin Ardüser[2], Salome Dürr 🔟[3], Stefanie Gobeli Brawand[4¤], Adrian Steiner[2], Patrik Zanolari[2], Marie-Pierre Ryser-Degiorgis 🔟[1] *

**1** Centre for Fish and Wildlife Health, Vetsuisse-Faculty, University of Bern, Bern, Switzerland, **2** Clinic for Ruminants, Vetsuisse-Faculty, University of Bern, Bern, Switzerland, **3** Veterinary Public Health Institute, Vetsuisse-Faculty, University of Bern, Liebefeld, Switzerland, **4** Institute of Veterinary Bacteriology, Vetsuisse-Faculty, University of Bern, Bern, Switzerland

¤ Current address: Federal Food Safety and Veterinary Office (FSVO), Bern, Switzerland
* marie-pierre.ryser@vetsuisse.unibe.ch

**Data Availability Statement:** All relevant data are within the manuscript and its Supporting Information files.

## Abstract

Footrot is a worldwide economically important, painful, contagious bacterial foot disease of domestic and wild ungulates caused by *Dichelobacter nodosus*. Benign and virulent strains have been identified in sheep presenting with mild and severe lesions, respectively. However, in Alpine ibex (*Capra ibex ibex*), both strains have been associated with severe lesions. Because the disease is widespread throughout sheep flocks in Switzerland, a nationwide footrot control program for sheep focusing on virulent strains shall soon be implemented. The aim of this cross-sectional study was to estimate the nationwide prevalence of both strain groups of *D. nodosus* in four wild indigenous ruminant species and to identify potential susceptible wildlife maintenance hosts that could be a reinfection source for domestic sheep. During two years (2017–2018), interdigital swabs of 1,821 wild indigenous ruminant species (Alpine ibex, Alpine chamois (*Rupicapra rupicapra*), roe deer (*Capreolus capreolus*), red deer (*Cervus elaphus*)) were analysed by Real-Time PCR. Furthermore, observed interspecies interactions were documented for each sample. Overall, we report a low prevalence of *D. nodosus* in all four indigenous wild ruminants, for both benign (1.97%, N = 36, of which 31 red deer) and virulent (0.05%, N = 1 ibex) strains. Footrot lesions were documented in one ibex with virulent strains, and in one ibex with benign strains. Interspecific interactions involving domestic livestock occurred mainly with cattle and sheep. In conclusion, the data suggest that wild ungulates are likely irrelevant for the maintenance and spread of *D. nodosus*. Furthermore, we add evidence that both *D. nodosus* strain types can be associated with severe disease in Alpine ibex. These data are crucial for the upcoming nationwide control program and reveal that wild ruminants should not be considered as a threat to footrot control in sheep in this context.

**Funding:** This study was financially supported by a grant of the Swiss Federal Food Safety and Veterinary Office (www.blv.admin.ch) together with the hunting authorities of the cantons of Fribourg, Grisons, Ticino, Schwyz and Nidwalden to MPRD and PZ (principal investigators), SD, SG, AS (grant Nr. 1.17.05). The funders had no role in study design, data collection and analysis, decision to publish, or preparation of the manuscript.

**Competing interests:** The authors have declared that no competing interests exist.

## Introduction

Footrot is an economically important, painful, contagious bacterial disease that affects feet of both domestic and wild ungulates [1,2]. Among domestic ruminants, mainly sheep are affected. Footrot is endemic in sheep flocks worldwide, including Switzerland [3]. In free-ranging wildlife, footrot has been documented in several European countries, affecting Alpine ibex (*Capra ibex ibex*) and mufflon (*Ovis orientalis orientalis*) [1,4–6] (Meneguz, Frey and Ryser, unpublished data). Since the treatment of wild ungulates in the field is challenging, expensive, and neither feasible nor desired by wildlife managers, severe lesions do not just affect animal welfare but typically result in death [4,6]. Considering that Alpine ibex in Switzerland have just recovered from the verge of extinction in the 20th century [7], outbreaks of footrot may be relevant to species conservation.

The main etiologic agent of footrot is *Dichelobacter nodosus* [8]. Lesions begin as a mild interdigital inflammation (mild form) and may progress to a severe interdigital ulceration up to the separation of the horn from the underlying skin (severe form) [9]. In sheep, mild and severe clinical forms of footrot have been shown to be associated with the presence of strains of *D. nodosus* carrying either the *AprB2* or *AprV2* genes, respectively, which encode for the subtilisin-like extracellular proteases playing a key role as virulence factors [8,10], thus referred to as "benign" and "virulent". However, in Alpine ibex, the benign strain was also found in association with severe lesions [4]. Furthermore, both types of strains of *D. nodosus* have been detected in the absence of lesions in sheep, cattle, goats and swine [11–14], suggesting the existence of species-specific differences in disease susceptibility and the involvement of other individual or environmental risk factors in the disease course. Among others, bacterial invasion is favored by interdigital skin damage caused by trauma or environmental conditions such as humid and wet pastures [15]. Unusually mild and wet weather conditions contribute to the persistence of *D. nodosus* in the environment [1,5]. In sheep, co-infection with anaerobic bacteria such as *Fusobacterium necrophorum* has been suggested to increase disease severity [9,11,16].

Due to an absence of long-term immunity, reinfections and clinical relapses are frequent in sheep [17]. Healthy carriers are known to occur in sheep, cattle and goats [11,12,14,18] and the bacteria can survive up to two weeks in the environment [11]. Virulent isolates may persist up to 10 months on bovine feet after co-grazing with sheep [12]. Furthermore, a single grazing season without contact with sheep is insufficient to eliminate all pathogenic isolates from cattle feet [12]. Thus, *D. nodosus* may be maintained in domestic ruminants even in the absence of clinical disease.

In the Swiss Alps, and to a lesser extent in the Jura Mountains, transhumance-grazing is a common practice and interactions between wild and domestic ruminants regularly occur on summer grazing pastures [19–21]. Pastures are often shared not only among cattle, sheep, goats and South American camelids (*Lama glama* and *Vicugna pacos*) [22], but also with indigenous free-ranging wild ruminants such as ibex (*Capra ibex ibex*), chamois (*Rupicapra rupicapra*), red deer (*Cervus elaphus*) and roe deer (*Capreolus capreolus*) [21]. All of the hoof stock represents potential hosts of *D. nodosus* and may play a role in pathogen maintenance and spread. Based on the widespread occurrence of the disease in sheep and its sporadic occurrence in ibex and mufflon, it has been postulated that sheep are the source of infection for wildlife, with transmission occurring on alpine pastures during the summer grazing season [1].

Currently, countrywide prevalence estimations for *D. nodosus* are available only for potential domestic hosts [14], while information on wildlife is virtually nonexistent. Importantly, further insight in the processes governing the epidemiology of *D. nodosus* infections in wild

and domestic ruminants and into the infection dynamics at the wildlife-livestock interface are needed to propose appropriate disease prevention and management measures in both domestic and wild animals. For this purpose, it is crucial that data obtained on wildlife and domestic animals can be directly compared [23], which requires the use of harmonized methods, from the sampling strategy to laboratory analyses [24]. Therefore, a nationwide survey on infections with *D. nodosus* in free-ranging ungulates was conducted, following the same study design and methodology as a parallel study in domestic livestock and South-American camelids in Switzerland [14].

The aim was to estimate the prevalence of *D. nodosus* infections, distinguishing between benign and virulent strains, in four species, namely Alpine ibex, Alpine chamois, roe deer and red deer during a two-year period (2017–18) in all of Switzerland. We hypothesized that wild ungulates do not maintain *D. nodosus* and that domestic ruminants act as the main infection source for wildlife. The obtained results enabled us to provide information on the likely role of the studied species in the epidemiology of footrot.

## Materials & methods

### Ethics statement

This study did not involve purposeful killing of animals. All samples originated from dead wildlife legally hunted during hunting seasons, found dead, or legally shot because of severe debilitation. According to the Swiss legislation (992.0 hunting law and 455 animal protection law, including the legislation on animal experiments; www.admin.ch), no ethical approval or permit for carcass /sample collection or animal experimentation was required.

### Study area, species of interest, study design and sampling strategy

The study area comprised the whole territory of Switzerland (41,285 km$^2$), which consists of three main bioregions: the Jura Mountains (a limestone mountain chain with an elevation up to 1,679m that separates the Alps to the southwest and forms an arc to the northeast), the Midlands (characterized by a low altitude and a high human population density), and the Alps (having the highest elevation up to 4,632m above sea level, which creates a climate wall separating the south from central Europe). Species of interest were indigenous free-ranging wild ruminants (roe deer, red deer, chamois and ibex). Deer species are mostly found below the timberline—roe deer mainly in the foothill zone and red deer in the mountain zone; chamois live between the subalpine and alpine zone; and ibex above the timberline.

The study was based on a cross-sectional convenient sampling strategy with the aim of estimating a nationwide prevalence of *D. nodosus* infections on an animal level in each of the four wild ruminant species. Sampling was carried out from August 2017 to December 2018, with most (89% N = 1622) samples collected during the hunting seasons (i.e. August-December) of 2017 and 2018.

Sample size was calculated for each species separately, assuming simple random sampling. The free online tool by AusVet Animal Health Services (http://epitools.auvet.com) was used for the calculation, and in there, the method for the estimation of true prevalence using imperfect diagnostic test characteristics was applied. For all species the design prevalence was set at 50% because no prior information on prevalence of infection was available. The precision was set at 5% and the level of confidence at 95%. The following estimated population sizes were used for the sample size calculation [25]: roe deer (113,000), red deer (28,500), chamois (91,500) and ibex (15,500). Diagnostic test characteristics (qPCR) were estimated to a mean sensitivity of 93.8% and a specificity of 98.3% [26]. We aimed to sample 440–451 animals per

species over two years, i.e. 1,786 individuals in total, distributed across the country and local political borders (cantons), according to species occurrence and hunting plans.

## Sample collection and animals

Sampling was mainly done by professional game wardens and hunters and in a few cases by the first and the second authors. Sampling kits included one SV Lysis buffer (4 M guanindinethiocyanate, 0.01 M Tris-HL, β-mercaptoethanol) in a tube with a screw-on lid, a cotton swab, a pair of latex gloves and a data sheet to record the sampling date, biological data of the animal sampled (sex, age, geographical origin, body condition) and the presence of foot lesions. Documentation of foot lesions were recorded for each foot separately, including lesions in the interdigital space, the feet and the carpal area based on the most common lesions that are used in footrot scoring systems [27]. If lesions were present, the sample submitters were asked to submit the affected feet in addition to the swabs.

The study participants were also asked to report previously observed interspecies interactions involving both wild and domestic species within a radius of 5 km around the sampling location. This distance was roughly estimated considering the home range size of the investigated species [28,29]. Four categories of contacts were used as previously described [21]: physical contact; encounter of less than 50 m distance between animals; encounters of more than 50 m; and non-simultaneous occupation of the same area. These categories were not relevant for *D. nodosus* transmission but corresponded to those used in earlier studies and helped specifying the notion of interaction for the respondents. Each data sheet included two tables, one for contacts among wild ruminants and the other one for contacts between wild and domestic ungulates. For each type of contact, the frequency of observation was recorded as follows: 0) never, 1) no more than once a year, 2) more than once a year.

All sampling kits were sent to the cantonal hunting authorities, which subsequently distributed the material to game wardens or hunters who then collected the samples from hunted game in the field. Animals that were sent as routine diagnostic cases (without footrot lesions) to the Centre for Fish and Wildlife Health (FIWI, University of Bern) for post-mortem examination were also sampled for this study (N = 54).

A four-feet sample was taken from each animal, i.e. the interdigital space of each of the four feet of each animal was sampled with the same cotton swab (2 mm 15 cm, Heinz Herenz, Medizinalbedarf GmbH, Hamburg, Germany). Each clean quarter of the swab was used for each foot as previously described [18,26], i.e. there was one swab per animal for laboratory analysis.

Immediately after swabbing the feet, each swab was immersed for at least 1 min in a tube containing SV lysis buffer with a hermetic lid. Samples were then either sent directly to the FIWI without cooling or stored at refrigerator temperature until transportation and subsequent analysis [10].

In total 1,821 samples were taken, of which 91.4% (N = 1,664) by game wardens/hunters; 4.5% (N = 83) by field biologists; 3.3% (N = 60) by veterinarians (FIWI) and for 0.7% (N = 14) of the samples the profession of the sampler was not given. The feet of five ibex, one red deer, three chamois and two roe deer with presumptive footrot lesions were sent to the FIWI for macroscopic examination. All 26 cantons contributed to the achieved sample number. 37.7% of the samples (N = 686) were received in 2017 and 62.3% (N = 1135) in 2018.

There were 961 males, 807 females and 53 animals without sex information. We used three age categories for the age estimation: juvenile (< 1 year, N = 167), yearling (1-<2 year; N = 245) and adult (≥ 2 years, N = 1,335). Age estimation was based on acquired knowledge of the sample submitters during their formal training (hunters and professional game wardens). For roe deer and red deer, this was based on the tooth replacement and wear, while for

ibex and chamois it was based on the growth rings of the horns [30]. No age information was provided for 74 animals.

## Laboratory analysis

All collected samples were tested for the presence of benign or/and virulent strains of *D. nodosus* by PCR [10]. Deoxyribonucleic acid (DNA) was extracted from the buffer solution by an automated method using a KingFisher™ Duo Prime purification automat (Thermo Scientific, Finland). A competitive real-time PCR was performed on all samples, which allows the simultaneous detection of virulent and benign strains of *D. nodosus* by distinguishing the extracellular protease AprV2 found in the more virulent strains from the subtly different protease AprB2 that is found in strains associated with benign disease signs in sheep [10]. Reagents used for the PCR included:TaqMan Fast Advanced Master Mix (Fisher Scientific, Reinach), Primers (Microsynth, AG Balgach), two probes Probe DnAprTM-v (Fisher Scientific) and Probe DnAprTM-b (Fisher Scientific), as well as a LIZ Assay Mix (Fisher Scientific). Amplification was done in a 7500 Real-Time PCR system with cycle conditions of 2 min at 50˚C, 10 min at 95˚C followed by 40 cycles of 15s at 95˚C and 1 min at 60˚C. Samples that showed no fluorescent signal were considered negative; samples showing a positive fluorescent signal for either strain were classified as positive. Results were then analyzed using the Sequence Detector 7500.

## Data management and statistical analysis

Data handling, validating, cleaning and coding were done in MS Excel spreadsheets followed by transfer to R software version 3.5.1. (https://cran.r-project.org) for all statistical analyses. Apparent and true prevalence were calculated using the "epi.prev" function of the package "epiR" considering imperfect test characteristics, with a test sensitivity of 93.8% and test specificity of 98.3% [26] and a confidence interval of 95%. The function calculates the true prevalence (TP; prevalence obtained from the apparent prevalence after correction and considering the non-perfect diagnostic test characteristics) from the apparent prevalence (AP; proportion of animals from a representative sample of the population that are positive to the diagnostic method used) using the Rogan-Gladen estimator, based in the formula $TP = (AP + specificity - 1)/(sensitivity + specificity - 1)$ [31].

A risk factor analysis for presence of *D. nodosus* on wild animal feet was performed with the PCR status of the animals (positive or negative) as dependent variable and potential risk factors as independent variables. Because only one sample was positive for virulent strains, the risk factor analysis was performed for benign strains only. The presence of foot lesions (presence *vs* absence) was not considered in the analysis because lesions were reported in only five PCR-positive animals and confirmed as footrot-like in only one ibex; the four red deer had either unconfirmed footrot-like lesions or merely overgrown hooves, which is not consistent with footrot. The factor "canton" was tested only for wild ungulates from cantons with PCR-positive animals because there were too many cantons without PCR-positive animals. Logistic regression models were used first in a univariable analysis, followed by a multivariable analysis. The variables interactions with either roe deer or with chamois were tested using Fisher's Exact Test because zero interactions were observed for PCR-positive animals. Variables with p-values < 0.2 were further considered in the multivariable analysis, where a manual backward elimination procedure was performed with a cut-off level at p-value <0.05. To be able to compare the prevalence for benign strains of *D. nodosus* in sheep and cattle (domestic ungulate study; Ardüser et al., 2019) with the prevalence of infection for benign strains in all wild

ungulates (all species pooled; this study), we used a logistic regression model in a multivariable analysis with a cut-off level at a p-value <0.05.

All maps were generated with QGIS 2.18.16 Las Palmas (Free Software Foundation Inc., Boston, USA).

## Results

### Detection of *Dichelobacter nodosus*

Out of 1,821 sampled animals (Fig 1), 37 were found positive for *D. nodosus* (Fig 2). Virulent strains were only found in one animal, an adult male ibex with severe disease signs (see also the section "Foot lesions" below). Benign strains were found in 36 animals, including 31 red deer (20 males and 11 females; 8 juveniles, 5 yearlings, 17 adults, and 1 animal of unknown

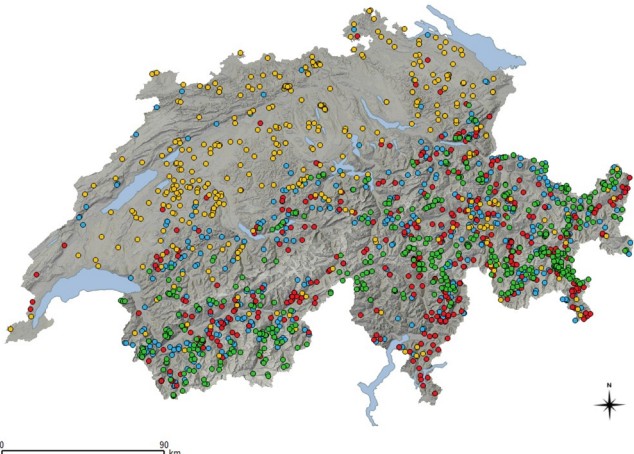

**Fig 1. Map of Switzerland showing the distribution of the sampled wild animals.** Shades of grey illustrate the mountainous shape. Lakes are in blue. Colored dots correspond to wild ungulates of different species: Yellow: roe deer; Red: red deer; Blue: chamois; and Green: ibex.

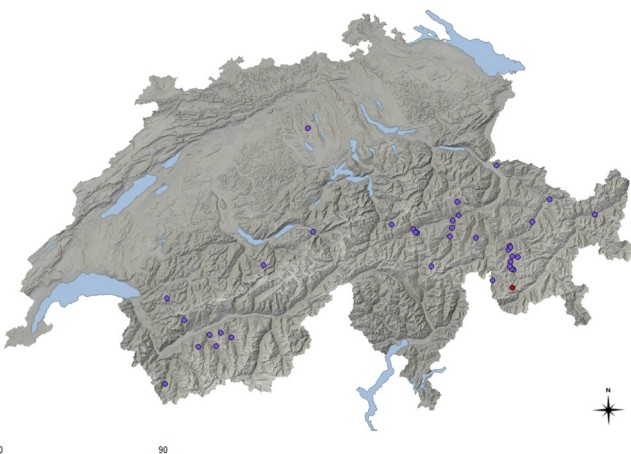

**Fig 2. Map of Switzerland showing the geographical origin of PCR positive animals.** Shades of grey illustrate the mountainous shape. Lakes are in blue. Violet dots correspond to wild ungulates positive for benign strains and red dots to wild ungulates positive for virulent strains.

**Table 1. Prevalence of *D. nodosus* in Swiss wild ungulates.**

| Species | Tested samples | Positive benign | Benign % AP | Positive virulent | Virulent % AP |
|---|---|---|---|---|---|
| Alpine ibex | 589 | 3 | 0.50% (0.13–1.46) | 1 | 0.16% (0.00–0.94) |
| Alpine chamois | 410 | 1 | 0.24% (0.00–1.35) | 0 | 0.00% (0.00–0.87) |
| Red deer | 408 | 31 | 7.59% (5.29–10.58) | 0 | 0.00% (0.00–0.88) |
| Roe deer | 409 | 1 | 0.24% (0.00–1.35) | 0 | 0.00% (0.00–0.87) |
| TOTAL | 1821* | 36 | 1.97% (1.39–2.72) | 1 | 0.05% (0.00–0.30) |

Number of sampled wild ungulates in all of Switzerland (August till December, 2017 and 2018), positive PCR results and estimated apparent prevalence (AP) with corresponding 95% confidence intervals indicated in parentheses.

*For five samples the species was not indicated by the submitter.

age), one roe deer (yearling female), one Alpine chamois (adult male) and three Alpine ibex (all adult males). The estimated apparent prevalence for the benign and virulent strains by species is given in Table 1 (wild animals) and Table 2 (domestic animals for comparison). Due to the low prevalence of *D. nodosus* per species, the true prevalence (TP) could only be calculated for red deer (TP = 6.08%, $CI_{95\%}$ = 3.5–9.3).

All positive samples originated from animals from six out of 26 Swiss cantons (S1 Table), all of them located in subalpine to alpine zones (> 1000 m above sea level) except for five red deer and one roe deer from the mountain and foothill zones. The majority (N = 27, 68%) of the positive animals originated from the canton of Grisons (23 red deer with benign strains; and two ibex: one positive for benign and the other one for virulent strains). Positive samples were found in both sampling years (14 in 2017; 23 in 2018). Most animals positive for benign *D. nodosus* (78%, N = 28) were reported to be in good body condition and a few animals were reported to be in moderate (19%, N = 7) and poor (3%, N = 1) body condition. The single ibex positive with the virulent strain was in a moderate body condition.

**Foot lesions.** Four of the 31 (12.9%) red deer positive for benign strains of *D. nodosus* were reported by the sample submitters as having foot lesions. Two of them were reported to

**Table 2. Prevalence of *D. nodosus* in Swiss domestic ungulates for comparison.**

| Species | Tested samples | Positive benign | Benign % AP | Positive virulent | Virulent % AP |
|---|---|---|---|---|---|
| Sheep | 690 | 58 | 7.80% (3.40–12.10) | 94 | 17.60% (10.70–24.40) |
| Cattle | 849 | 694 | 83.30% (79.10–87.50) | 0 | 0.00% (0.00–0.40) |
| Goats | 790 | 23 | 2.30% (0.00–5.10) | 0 | 0.00% (0.00–0.50) |
| SAC† | 591 | 13 | 0.90% (0.30–1.50) | 3 | 0.20% (0.00–0.40) |
| TOTAL | 2920 | 788 | 26.98% (23.38–28.62) | 97 | 3.21% (2.70–4.02) |

Ardüser et al., 2019

Number of sampled domestic ungulates in all of Switzerland (May 2017 till June 2018), positive PCR results and estimated apparent prevalence (AP) with corresponding 95% confidence intervals indicated in parentheses.

†South American Camelids

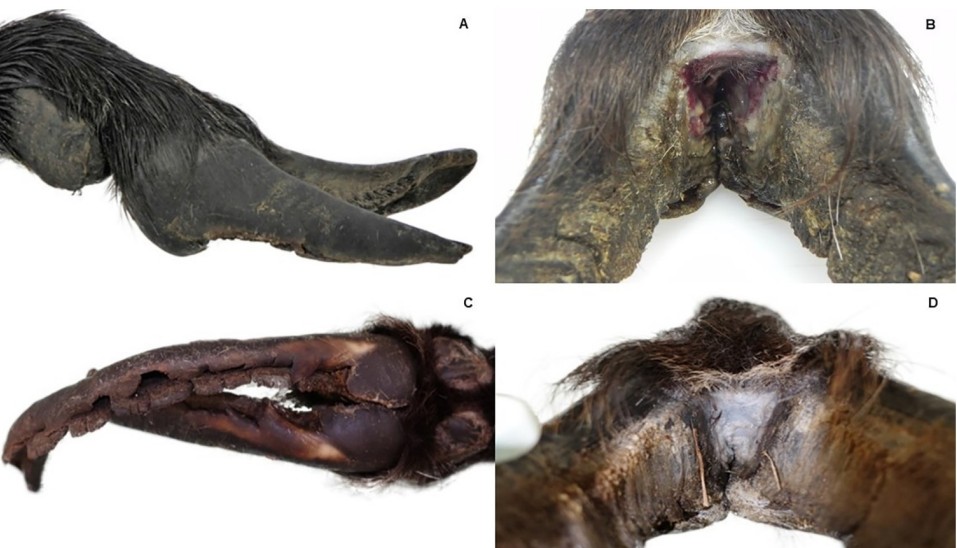

**Fig 3. *D. nodosus* positive feet of two ibex.** Front feet of an ibex, positive for virulent *D. nodosus*: (A) severely overgrown hooves with (B) ulceration and greyish discoloration of the interdigital space. Hind feet of an ibex, positive for benign *D. nodosus*: (C) greyish discoloration of the interdigital space with (D) severely overgrown and fissurated hooves.

present with changes consisting in overgrown hooves, one of which was submitted for veterinary examination. The anomaly concerned the lateral digits only, without noticeable interdigital inflammation (i.e. no signs suggestive of footrot). In the other two red deer, an interdigital grey malodorous exudate and rotting of the sole of one of the forelimbs was reported. However, these observations could not be confirmed as neither photographs nor any of the affected feet were submitted with the samples. One ibex positive for benign strains presented with severe footrot lesions of the hind feet (Fig 3). All other animals positive for benign strains were not reported to have any signs of disease (including two ibex).

The ibex with virulent *D. nodosus* showed severe footrot lesions in the front feet (Fig 3), including a severe ulcerative interdigital pododermatitis with greyish discoloration associated with an unpleasant odor, and severely multifocally fissured and overgrown hooves. The severity of the lesions were similar in the two ibex infected with benign and virulent strains, respectively.

**Interspecies interactions.** In total, 68% (N = 1236) of the field partners answered the questions listed on the data sheet regarding interspecies interactions in their sampling area, either partially (58%) or fully (10%). Data on reported proximity among wild and domestic ungulates (ibex, chamois, red deer, roe deer, sheep, cattle, goats and South American camelids) are summarized in Fig 4.

Encounters involving both wildlife and domestic ungulates were reported for all four investigated wild ungulates in all possible categories. Encounters of <50m, >50m and the non-simultaneous occupation of the same pasture were reported more frequently with cattle and sheep than with goats and South American camelids.

For ibex and chamois more specifically, about a third of the respondents reported encounters of <50m and >50m with sheep as well as with cattle in ibex. Regarding chamois only, half of the respondents reported encounters of <50m and >50m with cattle. As for the non-simultaneous occupation of pastures, 11% of the respondents mentioned the use of pastures in wildlife habitat by cattle and 8% by sheep. Physical contact between ibex and domestic ungulates

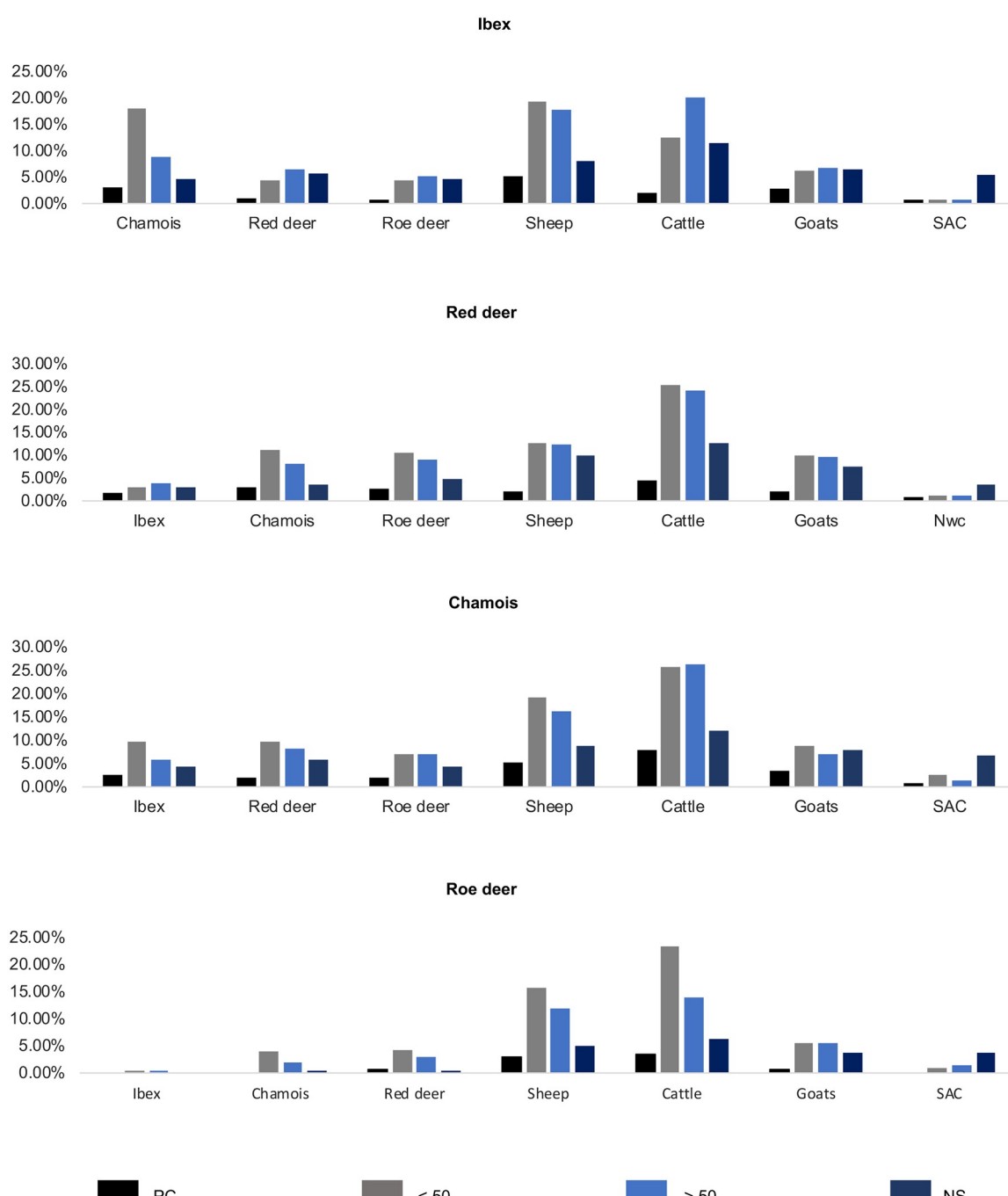

**Fig 4. Relative frequency of observations of different types of interspecies contacts between wild and domestic ruminants in Switzerland.** Proximity between species: 100% relates to all samples collected (information from respondents who did not report observations and lacking information due to non-reporting are not included in the graph). Colored bars indicate the following types of interactions: PC: physical contact; <50 m: encounter of <50 m; >50 m: encounter of >50 m; NS: non-simultaneous occupation of the same pasture.

was not frequent and reported mainly with sheep and goats (8% of the respondents). For chamois, a similar percentage of physical contact was reported with sheep and goats (9%), as well as with cattle (8%). With reference to red deer, encounters of <50m (25%) and >50m (24%) were observed with cattle, and 4% of the respondents even reported physical contact

between the two species. Regarding roe deer, patterns of encounters of <50m (23%) and >50m (14%) with cattle were similar to the observed interactions that were reported for red deer. The non-simultaneous occupation of the same pasture by deer and domestic ungulates was reported more frequently with cattle (red deer: 13%; roe deer: 6%) than with the other three domestic species.

Considering wild species only, interspecific interactions were reported mainly within the same taxonomic family, i.e. among caprids (ibex and chamois) and among cervids (roe deer and red deer). Interactions between red deer and chamois were also frequently reported and occasionally included physical contacts (5% of the respondents). By contrast, only a few interactions between ibex and cervids were reported, even though rare physical contacts between ibex and red deer (2%) were documented.

Focusing on positive samples, 83% of the submitters of the 28 red deer and the single roe deer samples positive for benign *D. nodosus* reported having observed direct or indirect contacts with deer and cattle in the sampling area. Regarding the single positive chamois, contacts among all wild and domestic ungulates, except domestic goats, were reported to have been observed in the sampling area. As concerns the four positive ibex, interspecies interactions were recorded only in the sampling area of two of them (both positive for the benign strain but without lesions), with all four domestic ungulates and with sheep only, respectively. As for the two ibex with severe footrot lesions, no information regarding interspecies interactions was provided.

**Analysis of risk factors for infection with *D. nodosus*.** For benign strains, univariable analysis indicated that the variables sex (male *vs* female), age (adult *vs* yearling and juvenile), species (ibex *vs* red deer, chamois and roe deer), and interspecies interactions in the sampling area (report *vs* no report) were somewhat associated with the detection of benign *D. nodosus* considering the threshold of p = 0.2 (Table 3).

All factors with a p value < 0.2, except "interspecies contacts" due to the large number of missing values, were additionally tested in a multivariable analysis, which revealed that the species "red deer" was a significant risk factor for *D. nodosus* carriage (Table 4). The model showed that red deer were 13.84 times more likely to be carriers of benign strains than ibex (CI: 4.70–59.08, estimate: 2.6277, standard error: 0.6230, z-value: 4.218, Pr (|z|) = < 0.001***).

In a further univariable analysis, the prevalence of infection in the four wild indigenous ruminants (all species pooled; this study) was compared to that found in sheep and cattle from the domestic ungulate study (Ardüser et al., 2019). The model revealed that compared to all indigenous wild ruminants both "cattle" and "sheep" were respectively 221.38 (95% CI 2.97–7.00) and 4.53 (95% CI 154.52–326.43) times more likely to carry benign strains of *D. nodosus*.

## Discussion

In this study, the prevalence of *D. nodosus* in wild ungulates in Switzerland was assessed for the first time, more specifically in all four indigenous wild ruminant species. These results provide a baseline necessary for the planning of a nationwide footrot control program in domestic livestock in Switzerland. Although the number of investigated animals was too low to provide strong data on a local level (e.g. cantonal), the sample size fulfilled the criteria for prevalence estimation per species on a country level. The occurrence of interspecies interactions among wild and domestic ruminants in an Alpine environment was additionally documented. Importantly, the study was conducted in parallel to a nationwide survey of *D. nodosus* in possibly susceptible domestic livestock species (May until June 2017–2018) [14]. The comparison of these data together with the reported interspecies interactions is of crucial importance in providing information on disease dynamics at the wildlife-livestock interface. [32].

**Table 3. Univariable risk factor analysis for presence of benign *Dichelobacter nodosus* on wild ruminant feet.**

| Variables | | Tested/total (%) | | p-value | OR$_{95\%}$ |
|---|---|---|---|---|---|
| | | positive for benign *D. nodosus* | negative for benign *D. nodosus* | | |
| Sex | Female | 12/807 (1.5%) | 795/807 (98.5%) | - | †Baseline |
| | Male | 24/961 (2.5%) | 937/961 (97.5%) | 0.138 | 1.69 (0.85–3.53) |
| Age | Juvenile | 8/167 (4.8%) | 159/167 (95.2%) | - | †Baseline |
| | Yearling | 6/245 (2.4%) | 239/245 (97.6%) | 0.205 | 0.49 (0.16–1.46) |
| | Adult | 21/1335 (1.6%) | 1314/1335 (98.4%) | 0.006 | 0.31 (0.14–0.77) |
| Species | Ibex | 3/589 (0.5%) | 586/589 (99.5%) | - | †Baseline |
| | Red deer | 31/408 (7.6%) | 377/408 (92.4%) | < 0.001 | 16.06 (5.68–67.24) |
| | Chamois | 1/410 (0.2%) | 409/410 (99.8%) | 0.523 | 0.47 (0.03–3.74) |
| | Roe deer | 1/409 (0.2%) | 408/409 (99.8%) | 0.524 | 0.47 (0.02–3.75) |
| ††Contacts | No report with cattle | 5/254 (1.6%) | 249/254 (98.4%) | - | †Baseline |
| | With cattle | 26/751 (3.5%) | 725/751 (96.5%) | 0.240 | 1.78 (0.73–5.32) |
| | No report with goats | 7/387 (1.6%) | 380/387 (98.4%) | - | †Baseline |
| | With goats | 10/322 (3.1%) | 312/322 (96.9%) | 0.267 | 1.73 (0.66–4.84) |
| | No report with ibex | 1/107 (0.9%) | 106/107 (99.1%) | - | †Baseline |
| | With ibex | 7/99 (7.0%) | 92/99 (93.0%) | 0.052 | 4.01 (0.70–75.46) |
| | No report with chamois | 0/68 (0.00%) | 68/68 (100%) | 0.123 | Inf. |
| | With chamois | 10/231 (4.3%) | 221/231 (95.7%) | | |
| | No report with roe deer | 0/68 (0.00%) | 68/68 (100%) | 0.066 | Inf. |
| | With roe deer | 10/189 (5.3%) | 179/189 (94.7%) | | |

† Level of comparison.

††The variable "contacts" refers to reported interspecies interactions with wild and domestic ungulates (all *D. nodosus* positive wild animal species pooled) within a radius of 5km around the sampling location.

### *Dichelobacter nodosus*-associated lesions

Two adult male Alpine ibex, the one infected with benign strains and the other with virulent strains, showed severe lesions consistent with footrot. These findings are in line with previous reports documenting that the presence of benign and virulent strains of *D. nodosus* can be associated with severe foot lesions in Alpine ibex that are comparable to typical lesions of footrot in sheep [1,4]. The classification into benign and virulent strains was developed in the framework of a study conducted in sheep [10,33] and may only be applicable to sheep. The occurrence of severe lesions associated with both groups of strains in ibex but not in other wild and domestic species suggests the existence of a species-specific difference in disease susceptibility, as it occurs for other pathogens such as *Mycoplasma conjunctivae* in wild (mostly with marked disease signs) *vs* domestic Caprinae (mostly without or with only mild signs) and *Brucella abortus* in North American elk (*Cervus canadensis*) and Bison (*Bison bison*)

**Table 4. Multivariable analysis of risk factors for infection with benign *Dichelobacter nodosus* in wild ungulates.**

| Variables | | p-value | OR$_{95\%}$ |
|---|---|---|---|
| Species | Ibex | †Baseline | - |
| | Red deer | <0.001 | 13.84 (4.70–59.08) |
| | Chamois | 0.514 | 0.47 (0.02–3.40) |
| | Roe deer | 0.463 | 0.46 (0.02–3.70) |

† Level of comparison.

(with signs) *vs* cattle (without signs) [4,34–36]. As for the other two Alpine ibex that were found positive for benign strains without reported foot lesions but were not sent for veterinary examination, it cannot be ruled out that they may have been in an early stage of infection. Nevertheless, it is not known whether infection with *D. nodosus* (benign and virulent) requires the contribution of other factors (e.g. severe traumatic foot lesions, reduced fitness of the animal, co-infection with other microorganisms) for lesions to develop.

Two red deer with reported foot lesions were tested positive for benign *D. nodosus.* Whether these lesions were footrot-like could not be confirmed because neither the affected feet nor photographs were submitted for veterinary evaluation. However, *D. nodosus* has already been isolated from sole ulcers in farmed red deer in New Zealand [35]. Although strain differentiation was not performed, it is likely that it was the benign type based on proteolytic effects observed in bacterial culture and because subsequent experimental infection of sheep resulted in the development of mild footrot lesions only. In cattle and free-ranging reindeer, foot diseases with similar clinical findings can involve pathogens such as *Fusobacterium necrophorum* (cattle and reindeer) and *Treponema* sp. (cattle) [37–39]. Infection with *Treponema* sp. was also recently shown to be associated with a dramatic outbreak of severe foot lesions including osteomyelitis in free-ranging North American elk (*Cervus canadensis*) [40–42]. Therefore, since all samples of the present study were tested via qPCR for the presence of *D. nodosus* only, we cannot exclude that other pathogens causing foot disease in domestic and wild ruminants may have been involved in the development of the observed lesions.

## Interspecies interactions

Since only 68% of the respondents answered (partially or fully) the questions regarding interspecies interactions on the data sheet, and because wildlife is most often active at dawn or dusk [7,43,44], it is likely that the frequency and intensity of reported interactions was underestimated. Furthermore, these data rely on reports of numerous voluntary participants and mainly on memorized events. Therefore, the information provided may be biased due to a lack of reporting or to misreporting. Nevertheless, the observers were wildlife professionals and this method represents the only feasible and efficient way of recruiting such data on a large geographical scale. Importantly, the reported interspecies interactions at the wildlife/domestic animal interface were in agreement with previous findings [20,21]. The most frequent interactions between wild and domestic ungulates were involving cattle or sheep. This is not surprising as these two livestock species are by far the most common ones on alpine pastures. Such data are useful to understand the epidemiology of infectious diseases in natural habitats, which are shared during certain periods by multiple species, wild and domestic.

## Identifying maintenance hosts

The present study revealed a very low prevalence of infection in all considered wild ungulate species. All prevalences calculated for wild species were significantly lower than those reported in both cattle (benign strains) and sheep (benign and virulent strains) [14]. There is no given threshold of prevalence to conclude that a pathogen is maintained in a population or not. Pathogen persistence in a population has been demonstrated to rely on the critical community size of a given population considering that the number of susceptible individuals does not drop under a certain threshold [45]. Nevertheless, in smaller wildlife populations, it has been previously reported that some infections agents can persist at low prevalence, provided that the pathogen can circulate between subpopulations of several species and that recurrent interspecific transmission occurs [34,46]. However, in the case of *D. nodosus* the difference in prevalence between wild and domestic ungulates is striking. Additional points to consider are: 1)

the endemic disease status of the Swiss sheep population [3,14]; 2) the recently gained knowledge on the widespread occurrence of *D. nodosus* in asymptomatic livestock in Switzerland (88% of cattle positive for benign strains) [3,14]; 3) the much larger sizes of cattle and sheep populations compared to wild ungulate populations [25,47]; 4) the more continuous distribution of domestic than wild animals over the country (ibex in particular occur in fairly isolated colonies [25,41]; 5) the more frequent and intense intraspecific contacts among domestic animals and mixing of herds (grazing, shows, markets) and movements throughout the country (alpine summering, commercial exchanges); 6) the absence of healthy carriers for the virulent strain in wild ungulates; and 7) the reported interspecies interactions at the domestic animal/ wildlife interface regarding mainly sheep and cattle. Therefore, taking into consideration the previously mentioned points together with the comparison of the wild and domestic ungulate study, the results suggest that sheep are maintenance hosts for both virulent and benign *D. nodosus* and cattle for benign strains only. Thus, both may represent the main source of infection for domestic and wild ungulates. By contrast, our results suggest that wild ruminants are occasional spillover hosts without epidemiological significance. Nevertheless, the sample size used in this study does not allow to exclude potential differences in prevalence of infection on a local level. For example, the occurrence of previous outbreaks of footrot in wildlife, wildlife population management and densities, frequency and intensity of interspecific interactions on summer pastures and livestock practices such as already existing sanitation programs (for virulent strains) are all factors that likely vary among regions.

Regarding benign strains, infections were detected in at least one animal per wild ungulate species, with red deer showing the highest prevalence of infection (TP 6.08%). Risk factor analysis indicated that red deer are at a significantly higher risk of being carriers of benign *D. nodosus* than ibex, chamois and roe deer. This is in line with the fact that interspecies interactions involving red deer are most commonly observed with cattle (this study; 20) and that the prevalence of benign *D. nodosus* is highest in cattle (AP 83.3%; [14]. Therefore, cattle may act as source of infection for susceptible wild ungulates, as previously suspected [4]. Nevertheless, further molecular studies are needed to univocally prove that interspecies transmission of *D. nodosus* occurs between wildlife and domestic livestock. Since the upcoming nationwide control program will only focus on virulent *D. nodosus* in sheep, it is unclear whether it will influence the prevalence of benign *D. nodosus* in any species, i.e. the control program is not expected to significantly decrease the risk of footrot outbreaks caused by the benign strain of *D. nodosus* in ibex populations in the future.

## Conclusion

This study delivers crucial information for the design of the upcoming nationwide control program of virulent *D. nodosus* in sheep. Furthermore, the simultaneous, harmonized investigation of both wild and domestic species contributes to a better understanding of the epidemiology of footrot in Switzerland. This will help targeting disease prevention measures, as it revealed that combatting virulent strains does not need specific wildlife management measures. The obtained data suggest that infections in wild ruminants are sporadic despite the high prevalence of infection in sheep and cattle, and despite the widespread occurrence of interactions between wildlife and livestock. Overall, wildlife seems to be an incidental spillover host and not a maintenance host that may infect healthy sheep or re-infect sanitized herds. Since footrot lesions in ibex were also associated with benign strains and the upcoming control program will only focus on virulent strains in sheep, this program is expected to only partially influence disease occurrence in ibex in the future, especially when taking into consideration the high prevalence of benign *D. nodosus* in the Swiss cattle population. Further research is

needed to evaluate the impact of footrot in ibex and assess whether it would be appropriate to take disease management actions to prevent outbreaks.

## Supporting information

**S1 Table. Cantons with PCR positive animals.** Estimated true**/apparent* prevalence of *D. nodosus*, with corresponding 95% confidence intervals indicated in parentheses and number of tested animals.
(PDF)

## Acknowledgments

We are grateful to all game wardens, hunters and cantonal hunting inspectors who contributed to sample collection. We also thank all students and FIWI collaborators, especially Stefania Vannetti and Simone Pisano, for their contributions to sample collection and processing. Many thanks go to the staff of the Institute of Bacteriology, especially Simon Feyer and Anita Jaussi, for their contributions to laboratory analyses.

## Author Contributions

**Conceptualization:** Salome Dürr, Stefanie Gobeli Brawand, Adrian Steiner, Patrik Zanolari, Marie-Pierre Ryser-Degiorgis.

**Formal analysis:** Gaia Moore-Jones.

**Funding acquisition:** Salome Dürr, Stefanie Gobeli Brawand, Adrian Steiner, Patrik Zanolari, Marie-Pierre Ryser-Degiorgis.

**Investigation:** Gaia Moore-Jones, Flurin Ardüser, Stefanie Gobeli Brawand.

**Methodology:** Salome Dürr, Stefanie Gobeli Brawand, Patrik Zanolari, Marie-Pierre Ryser-Degiorgis.

**Project administration:** Patrik Zanolari, Marie-Pierre Ryser-Degiorgis.

**Supervision:** Salome Dürr, Marie-Pierre Ryser-Degiorgis.

**Writing – original draft:** Gaia Moore-Jones, Marie-Pierre Ryser-Degiorgis.

**Writing – review & editing:** Flurin Ardüser, Salome Dürr, Stefanie Gobeli Brawand, Adrian Steiner, Patrik Zanolari.

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
