## [Decision Letter · Decision Letter 0]

9 Aug 2019

PONE-D-19-18147

Identifying maintenance hosts and risk factors for infection with Dichelobacter nodosus in free-ranging wild ruminants in Switzerland: a prevalence study

PLOS ONE

Dear Prof. Ryser-Degiorgis,

Thank you for submitting your manuscript to PLOS ONE. After careful consideration, we feel that it has merit but does not fully meet PLOS ONE’s publication criteria as it currently stands. Therefore, we invite you to submit a revised version of the manuscript that addresses the points raised during the review process.

Many thanks for submitting your interesting article to PLOS One

It was reviewed by two expert reviewers, and major revisions were recommended

Both reviewers made some very useful comments to aid revision

Please provide a detailed rebuttal to reviewers comments and then the revised manuscript will be sent back to the same two reviewers

Wishing you the best of luck with your modifications

Many thanks

Simon

We would appreciate receiving your revised manuscript by Sep 23 2019 11:59PM. To enhance the reproducibility of your results, we recommend that if applicable you deposit your laboratory protocols in protocols.io, where a protocol can be assigned its own identifier (DOI) such that it can be cited independently in the future. For instructions see: http://journals.plos.org/plosone/s/submission-guidelines#loc-laboratory-protocols

We look forward to receiving your revised manuscript.

Kind regards,

Simon Russell Clegg, PhD

Academic Editor

PLOS ONE

Journal Requirements:

Reviewers' comments:

Reviewer's Responses to Questions

**Comments to the Author**

1. Is the manuscript technically sound, and do the data support the conclusions?

Reviewer #1: Yes

Reviewer #2: Yes

2. Has the statistical analysis been performed appropriately and rigorously? 

Reviewer #1: Yes

Reviewer #2: Yes

3. Have the authors made all data underlying the findings in their manuscript fully available?

Reviewer #1: Yes

Reviewer #2: Yes

4. Is the manuscript presented in an intelligible fashion and written in standard English?

Reviewer #1: Yes

Reviewer #2: Yes

5. Review Comments to the Author

Reviewer #1: This is a very interesting, well written paper which investigates the risk of the footrot bacteria, Dichelobacter nodosus on wild populations of ruminants in Switzerland. Using PCR assays, the authors show a low level of infection, and link this to a previous study of farmed animals.

Although the paper is very well written, I have a few minor comments, mainly grammatical, and a few questions which I thought of during reading the paper.

None of my comments are particularly major, nor do I feel too strongly over any, so if you disagree with any of them, then please just say.

Abstract

Line 17- D. nodosus in brackets isn’t needed. This is common bacterial nomenclature

Line 19, comma after (Capra ibex)

Line 19- fatal lesions? Can you be sure that the foot lesions killed the animal? Maybe remove it as this sounds a bit strange.

Line 19, add a comma after ‘In Switzerland’

You also refer to the benign and virulent strain throughout. It is my understanding that there are a variety of subtypes/ strains which are virulent, and others which are benign. Maybe this need some clarification within the document?

Introduction

Line 42-44. You discuss previous infection of other free ranging ruminants. It may be useful to state where in the world this was, as if it was local to Switzerland, then it may be beneficial to your study.

Line 44-46- I presume that the lame animals would be subject to increased predation? This maybe worth discussing as I am not sure what predators are seen in Switzerland. This may remove the number of lame animals by predation.

Line 49- D. nodosus in brackets isn’t needed. This is common bacterial nomenclature

Line 52. Comma needed between ibex and the

Line 56 ‘animal-individual’ is a bit unclear. Maybe you could reword it?

Line 74-75 – Sporadic character sounds slightly unusual. Maybe consider rewording?

Line 90 It may be helpful to state the four different wild ruminants here.

Line 92-94- not sure that this is needed.

Materials and methods

Line 56- Make it materials and methods

Line 99- You mention the use of dead animals. Would you know how long they had been dead for? I only ask as this may affect colonisation of the feet by the pathogens. If you have the data for the dead animals, would it be possible to analyse the prevalence in these animals to ascertain if death is an issue in skewing the results? I doubt it will be, but worth a look.

Line 100- You use the phrase including roe deer, red deer etc…. In reality I think this isn’t including, it is limited to as you only test the 4 different wild animals

Lines 155-157. This is a methodology I have not encountered before in my footrot work. Is there a risk that you are diluting out the bacteria on a swab from foot one by rubbing it onto other feet? Just wondered how accurate that method was? (doesn’t mean a change on the manuscript, more a question out of interest)

And did this methodology change when a lesion was seen to prevent spreading the bacteria, or equally to prevent losing it?

Line 160. You mention that it involves storage and transportation without cooling? How does that work when it is only put into a lysis buffer? Is it that you try to keep it temperature stable?

Lines 162- 173 I would say should be in the results section

Line 178- is it possible to state the PCR reagents so anyone else could follow it?

Line 190-191- You mention the apparent and true prevalence. Could you please define what these are?

Lines 194-195- the equation looks to be in a different type font

Results

Line 213- you mention severe clinical signs. Could you mention what these were? Or direct the reader to where they are mentioned as I believe they are mentioned further on in the document.

Lines 213-216. Again, it would be nice to know if any clinical signs were seen here, or point the reader to where this is discussed

Figure 1 legend. Line 222. I am not sure what you mean by the relief?

Within the results there are a lot of images and tables which are very close together. Some narrative between these may make it easier to follow (although this may change with editing).

In table 1a, the virulent column doesn’t really say much as many of the results are negative. Is this column worthwhile? Happy to be led by you, if you feel that you want it to stay then I wont argue with you.

The legends for table 1a and 1b would benefit from a bit more detail as to what the tables show. Also please define what SAC means in table 1b.

Line 241. A comma between S1 table) and all may aid flow here

Line 249, again, a comma between red deer and positive would aid flow.

Line 256, benign strain presented with severe

Line 264. Front feet of an ibex which was/ were positive. Alternatively a comma would suffice

And line 266- hind feet of an ibex which were positive. Alternatively a comma would suffice

Line 269- You discuss interspecies interactions. How clear would this be? Is it obvious and noticeable when they interact, or is it more that the animals were found in the same area? I think it is very interesting, but just want to try and understand it a little more.

Line 282 and 283. You mention small domestic ruminants. Do you mean cattle and sheep here? Or others too? Could you be more specific?

Line 283-284- As concerns seems slightly strange wording. Maybe with reference to?

Line 288. You switch her to domestic ungulates? Could you stick to the same terms as used previously, and maybe specific sheep and cattle, or others. Or even a link to table 1b may be useful?

Line 293- physical contacts. How do you define this?

Line 299. A pair of commas around except domestic goats, maybe help flow here

Line 317- what made you choose a cut off value of 0.2?

Table 2. If you could merge the top two lines where possible it would make the table clearer

Also it is clearer and more helpful not to leave gaps in the table, maybe an N/A would be more beneficial?

Line 330, you mentioned the species red deer- is that being of that species or encountering it?

Table 3. Maybe discuss in the legend what you mean by baseline?

Discussion

Line 256. A full stop is needed between livestock species, and The comparison

Line 361- Insertion of strain after the word benign may aid reading

Line 372- ‘As for the other two Alpine ibex that harboured the benign strain’ This sounds a bit colloquial. Maybe consider rewording.

Lines 382-389- you discuss the potential involvement of Treponema species. The academic editor handling this manuscript is an expert in Treponema species, so it maybe worth a discussion with him to see if he would be willing to run the samples for you as a collaboration to further this work? Just a thought

Line 400- maybe be biased? But how?

Line 400. You discuss the interactions, but would the presence of people (possibly hunters) affect this?

Line 404. You talk about interactions mainly being with cattle and sheep- but is this not due to the fact that these are the most abundant animals?

Line 412-415- I read this a few times, but never fully understood what you were trying to say. Maybe consider rewording it?

Line 419- comma after nodosus would aid reading flow.

Line 434- avoid the use of our.

Line 457- does not necessitate to involve wildlife doesn’t seem to make sense, and is unclear- consider rewording

Line 458- you use prevalence previously to be plural, so I would avoid prevalence’s

Line 461- what do you mean by a sanitised herd?

Figure 1 and 2. Would it be possible to combine these? Possibly by using a thicker black border around the circles which appear on both figures?

The graphs are a little unclear- but that may just be my print out

I hope my comments are useful and helpful for you.

Many thanks for the opportunity to review this paper

Reviewer #2: A manuscript entitled “Identifying maintenance hosts and risk factors for infection with Dichelobacter nodosus in free-ranging wild ruminants in Switzerland: a prevalence study” had the goal of identifying cases of benign and virulent D. nodosus in common wild ungulates.

Strengths: The authors report the PCR results of 1821 sampled wild ungulates collected between 2017-2018, which is an impressive number of animals representing the most common wild ungulates with ranges covering most of Switzerland, and in a short enough time period, such that results likely reflect the general overall trend of this organism. The number of species sampled were mostly balanced. The method of foot swabbing, field preservation and D. nodosus PCR has been well elaborated in previous publications and was likely effective. Authors were able to deduce that prevalence of D. nodosus was very low in wild ungulates despite common wild -domestic ungulate interface in most of the regions sampled. This is novel and valuable information and timely as Switzerland moves to report and control virulent D. nodosus in domestic livestock. This work was done in parallel with similar published work on domestic livestock, which strongly demonstrates that domestic livestock prevalence of D. nodosus is common and significant in comparison to wild ungulates.

Weaknesses:

• The title suggests that “risk factors” for Footrot in non-domestic ungulates will be evaluated, though risk factors are only loosely suggested in the text with no data in the study design to make an observation or conclusion as to risk. If this is meant to broadly encompass domestic-wild population interface, this should be worded differently.

• This manuscript relies heavily on observations and sample collection using untrained participants. This is worrisome for accuracy of specimens (were negative swabs truly negative cases?) and overall identification and interpretation of lesions, especially early lesions. This is particularly of concern when detailing wild – domestic animal interface, which was determined entirely from historic memory of untrained participants. The weaknesses of this data are well detailed by the authors in a paragraph starting Ln 395, and the authors should be careful to draw conclusions about transmission between livestock and wildlife, Ln 446. Further molecular work may need to be done comparing ibex and cattle / sheep isolates to make these conclusions.

• Authors attempt to correlate gross and lack of gross lesions with PCR results in wild ungulates but rely mostly on participants reports and a very small subset of feet submitted for professional evaluation. This limitation should be considered throughout the text when discussing the relevance of species contact.

• Discussion of pathogenesis and progression of foot lesions is not appropriate from the small subset of feet attained and is also not within the stated scope of the manuscript. The gross ibex foot lesions demonstrated are impressive and similar to that described by Wimmershoff et al 2015, but are these caused by D. nodosus? Histopathology, a larger subset of hooves representing various degrees of disease, along with a more detailed analysis of associated microorganisms would be necessary to call these primary D. nodosus lesions. D. nodosus associated? Please keep this deficit of information in mind when discussing disease development.

• There is no mention of season of data collection and particularly seasonal correlation with positive D. nodosus cases in wildlife, though this would likely have an influence in disease occurrence and progression. Could this be determined from the data?

• This study was done in parallel with a similar published study in domestic livestock. Where there is spacial overlap of wild D. nodosus cases with positive domestic cases, would it be possible to perform more detailed molecular work (sequencing, etc.) comparing the wild and domestic D. nodosus to demonstrate or exclude the possibility of a shared microorganism passing from domestic animals to wildlife?

Below are addition comments:

Ln 16 Footrot is a disease of feet and not specifically the hoof. Unless the lesion is being described only within the structures of the hoof, these references should be changed to foot / feet.

Ln 18, 51, 256 Please add …presenting “with” mild and …..

Ln 19 The sentence starting “In Switzerland..” seems out of place, as no correlation is established in the abstract between Ibex and domestic sheep. The abstract needs to be organized so that this information falls into logical place and leads to why wild ruminants are to be examined.

Ln 21 Please clearly note if “maintenance hosts” are wild or domestic or both, and if “sheep” are domestic or wild.

Ln 31 This appears a big jump to implement domestic animals as the cause of disease in wild animals. That was not the focus of the study and appears to be something that would need to be proven under controlled lab conditions and with specific molecular work to prove identical type stains. Noting that there is a low incidence of wild carriers and results in severe disease in wild hoofstock appears more in line with the aims and outline work, at least in the abstract.

Ln 45 Please change to “practical”

Ln 51 Introduce the molecular or other difference between “benign” and “virulent” strains here. How do you differentiate these organisms.

Ln 56 How does co-infection play into D. nodosus infection and susceptibility in domestic animals? This would be appropriate to discuss here.

Ln 72 Replace “of them” with “hoof stock”

Ln 131 Do the authors have a sense of reproducibility and agreement on lesion scoring, description and swabbing technique between game wardens, hunters, etc.? If so, please note how that agreement was determined. My experience is that there is a very big difference between lesion grading done by biologists, lay people and veterinarians and particularly with interpretation of early lesions. Were wardens and hunters trained in appropriate swabbing and handling techniques? If not, could this have confounded the findings? Explain.

Ln 134 Was season taken into account for sampled collected? Was this information recorded? It seems environmental conditions would play importantly into risk of foot rot and general overall health of the animal.

Ln 138 How were affected feet submitted? What was time elapsed between collection and receipt in the lab and how were the feet handed / analyzed once in the lab?

Ln 139 Is this a historical report generated by the participant? Over what period of time are these observations of a 5 km radius representing? At the time of animal collection? Over what season?

Ln 229, 237 These table legends should stand alone and clearly delineate these as Swiss wildlife and domestic ungulates, and the period for which the data was collected. Are these domestic ungulate results taken in the same time interval as wildlife? This will be important to state or explain how different time points are comparable.

Ln 256 Were gross feet available to the authors in all cases with positive D. nodosus PCR? Or is this statement based on participant reports?

Ln 263 Were cases only evaluated by PCR for D. nodosus with gross examination of a small subset of cases with lesions?

Ln 264 These are impressive gross lesions. For the outlined study, finding of D. nodosus and specifically the virulent strain is important, but what other organisms are associated with this lesion? Treponema? Fusobacterium? Most foot diseases are far more complex than a single microorganism. Perhaps beyond the scope of this article, but it would be nice to see representative sections of this lesion histologically and perhaps demonstrating the microorganism and any associated pathogens. What time of year was this animal collected? Were there other conditions found on processing the cadavers?

Ln 269 The observations by participants is helpful, but could there not be structured information by trained wardens / biologists to more objectively report wild and domestic animal interface in these cantons? Clarify here as well, during what time period and season are these observations taken? Make sure that abbreviations are decoded in a foot note on all figures and tables and that titles and legends are descriptive sufficiently so that the items can stand alone when read separate from the text.

Ln 386 This citation should include Treponema associated hoof disease in free-ranging elk Vet Pathol 2019.

Ln 390 Since this work evaluated a one-time PCR for D. nodosus with no controlled gross observations and missing data on subjects, it does not seem appropriate to discuss pathogenesis and progression of D. nodosus lesions in ibex.

6. PLOS authors have the option to publish the peer review history of their article (what does this mean?). If published, this will include your full peer review and any attached files.

Reviewer #1: No

Reviewer #2: No

---

## [Author Response · Author response to Decision Letter 0]

27 Sep 2019

We sincerely thank the two reviewers for their thorough assessment of our manuscript. We have carefully considered all their comments and believe that the manuscript has improved. We would like to point out that the answers to the reviewer's comments refer to the manuscript without track changes.

Reviewer #1: 

This is a very interesting, well written paper which investigates the risk of the footrot bacteria, Dichelobacter nodosus on wild populations of ruminants in Switzerland. Using PCR assays, the authors show a low level of infection, and link this to a previous study of farmed animals.

Although the paper is very well written, I have a few minor comments, mainly grammatical, and a few questions which I thought of during reading the paper.

None of my comments are particularly major, nor do I feel too strongly over any, so if you disagree with any of them, then please just say.

I hope my comments are useful and helpful for you.

Many thanks for the opportunity to review this paper.

1) Line 17- D. nodosus in brackets isn’t needed. This is common bacterial nomenclature

Ans: Done.

2) Line 19, comma after (Capra ibex)

Ans: Done.

3) Line 19- fatal lesions? Can you be sure that the foot lesions killed the animal? Maybe remove it as 

this sounds a bit strange.

Ans: Field observations have shown that impairment in movement in these free ranging wild ruminants means that their spatial movement is strongly reduced as they end moving on their carpal joints or stay still. Thus, their foraging abilities are reduced as well. If they are not found dead (e.g. in avalanches), they are mostly shot by game wardens for animal welfare reasons. Therefore, we think that underlying the fact that these lesions have serious consequences is important but we have re-worded the sentence.

4) Line 19, add a comma after ‘In Switzerland’

Ans: Done.

5) You also refer to the benign and virulent strain throughout. It is my understanding that there are a variety of subtypes/ strains which are virulent, and others which are benign. Maybe this need some clarification within the document?

We have added extra clarification in the introduction .

Line 48-52 " In sheep, mild and severe clinical forms of footrot have been shown to be associated with the presence of strains of D. nodosus carrying either the AprB2 or AprV2 genes, respectively, which encode for the subtilisin-like extracellular proteases playing a key role as virulence factors (8,10), thus referred to as “benign” and “virulent”. Indeed, in both cases (mild and severe), it is a group of strains and not a single strain each. " We adapted the manuscript as appropriate, i.e. replacing “strain” by “strains” or “strain group/type”.

6) Line 42-44. You discuss previous infection of other free ranging ruminants. It may be useful to state where in the world this was, as if it was local to Switzerland, then it may be beneficial to your study.

Ans: The manuscript was adapted accordingly. 

7) Line 44-46- I presume that the lame animals would be subject to increased predation? This maybe worth discussing as I am not sure what predators are seen in Switzerland. This may remove the number of lame animals by predation.

Ans: So far, predation on ibex in Switzerland has been limited to juveniles attacked by eagles or lynx. Of course, it may well be that not all animals with lesions were found. This issue is inherent in all investigations on free-ranging wildlife. Nevertheless, the main population of interest for this prevalence study consisted in non-affected animals (potential healthy carriers). 

8) Line 49- D. nodosus in brackets isn’t needed. This is common bacterial nomenclature

Ans: Done.

9) Line 52. Comma needed between ibex and the

Ans: Done.

10) Line 56 ‘animal-individual’ is a bit unclear. Maybe you could reword it?

Ans: Done.

11) Line 74-75 – Sporadic character sounds slightly unusual. Maybe consider rewording?

Ans: Done ("sporadic occurrence" instead of sporadic character)

12) Line 90 It may be helpful to state the four different wild ruminants here.

Ans: Done.

13) Line 92-94- not sure that this is needed.

Ans: Since the other reviewer did not suggest removing this section, we would prefer to keep our work hypothesis here.

14) Line 56- Make it materials and methods

Ans: Done.

15) Line 99- You mention the use of dead animals. Would you know how long they had been dead for? I only ask as this may affect colonisation of the feet by the pathogens. If you have the data for the dead animals, would it be possible to analyse the prevalence in these animals to ascertain if death is an issue in skewing the results? I doubt it will be, but worth a look.

Ans: Regarding animals found dead, it would not be possible to find out how long the animals have been dead for. This may have affected D. nodosus survival but as PCR detect bacterial DNA (even if bacteria are dead), this should not have affected the PCR results. Hunted animals (the large majority of our samples) were sampled right after death. We verified the difference in prevalence between animals found dead and animals that were shot. Out of 1821 sampled animals, 1632 (89%) were shot and 121 (6.6 %) were found dead. For 68 (4.4%) animals, this information was not given. A comparison was not possible in ibex as only 3 animals were positive for the benign strain (all shot). A potential difference could only be tested for red deer, in which the prevalence of D. nodosus was 15.4% (2/13) in animals found dead and 7.7% (29/376) in animals that were shot. No statistical significant difference was found (p value = 0.327; CI =2. (0.3- 8.6) between these two groups. 

16) Line 100- You use the phrase including roe deer, red deer etc…. In reality I think this isn’t including, it is limited to as you only test the 4 different wild animals

Ans: We reworded the sentence as follows: 

Line 112-113: "Species of interest were indigenous free-ranging wild ruminants (roe deer, red deer, chamois and ibex)."

17) Lines 155-157. This is a methodology I have not encountered before in my footrot work. Is there a risk that you are diluting out the bacteria on a swab from foot one by rubbing it onto other feet? Just wondered how accurate that method was? (doesn’t mean a change on the manuscript, more a question out of interest)

And did this methodology change when a lesion was seen to prevent spreading the bacteria, or equally to prevent losing it?

Ans: The methodology we used has been previously described in a study conducted in Switzerland (Greber et. Al 2018, Pooling of interdigital swab samples for PCR detection of virulent D. nodosus). This study compared sampling individual feet to a 4-feet sample as done in our study. The results of Greber et al. show that with the 4-feet sample the sensitivity and specificity for the detection of aprV2 were 93.8% and 98.3, respectively. Wildlife samples were taken on dead animals only (found dead or shot), therefore a potential spreading of the bacteria during sampling was irrelevant. A loss of bacteria was possible but instructions were provided to sample the healthy feet first and since animals with observed footrot like lesions were all PCR-positive, the sampling procedure does not seem to have interfered with the results.

18) Line 160. You mention that it involves storage and transportation without cooling? How does that work when it is only put into a lysis buffer? Is it that you try to keep it temperature stable?

Ans: We used the method described in a previous study conducted at our Faculty (Stäuble et. al 2014; Molecular genetic analysis of Dichelobacter nodosus proteases AprV2/B2, AprV5/B5 and BprV/B in clinical material from European sheep flocks). Before shipment of the material to the field partners, the tubes with lysis buffer were kept away from sunlight and the sampling protocol stated that the samples should be sent back no later than 2 weeks after sampling, and that until then the samples should be stored at refrigerator temperature. Transportation back to our laboratory was done without cooling, since components of the buffer solution allow for a stabilization of the DNA without cooling requirement. 

19) Lines 162- 173 I would say should be in the results section

Ans: We prefer to leave this text in the Materials and Methods section since this was the material collected for testing, and not results of analyzes. Since the other reviewer did not have any objections to where this section is placed, we left the text as it was.

20) Line 178- is it possible to state the PCR reagents so anyone else could follow it?

Ans: We chose not to list all the reagents needed for the PCR to avoid the manuscript to become lengthy and distract the reader from the focus of the article. Detailed information can be found in the paper of Stäuble, et. al (2014, Molecular genetic analysis of Dichelobacter nodosus proteases AprV2/B2, AprV5/B5 and BprV/B in clinical material from European sheep flocks. Veterinary Microbiology) which is indicated as a reference and lists all the details concerning the molecular biological procedures needed to perform the Real Time PCR to detect D. nodosus. 

21) Line 190-191- You mention the apparent and true prevalence. Could you please define what these are?

Ans:

1) The apparent prevalence is the proportion of animals from a representative sample of the population that are positive to the diagnostic method used. It does not consider non-perfect test characteristics (i.e. sensitivity and specificity below 100%). 

2) The true prevalence is the prevalence obtained from the apparent prevalence after correction considering the non-perfect diagnostic test characteristics (sensitivity and specificity). 

We furthermore added a short definition of the AP and TP in the manuscript. 

22) Lines 194-195- the equation looks to be in a different type font

Ans: Done.

23) Line 213- you mention severe clinical signs. Could you mention what these were? Or direct the reader to where they are mentioned as I believe they are mentioned further on in the document.

We added in the introduction a specification as to what are mild and severe lesions: 

Line 46-48 " Lesions begin as a mild interdigital inflammation (mild form) and may progress to a severe interdigital ulceration up to the separation of the horn from the underlying skin (severe form).

24) Lines 213-216. Again, it would be nice to know if any clinical signs were seen here, or point the reader to where this is discussed

Ans: No particular clinical signs were reported by the submitters, foot lesions of D. nodosus positive animals are described in the results section "foot lesions". We have added a specification in the manuscript for further clarification. 

Line 223-225: " Virulent strains were only found in one animal, an adult male ibex with severe disease signs (see also the section "Foot lesions" below)."

25) Figure 1 legend. Line 222. I am not sure what you mean by the relief?

Ans: “Relief” has several meanings. We meant “raising shapes above a flat surface so that they appear to stand out slightly from it”. For more clarity, we reworded relief with "mountainous shape" throughout the whole manuscript.

26) Within the results there are a lot of images and tables which are very close together. Some narrative between these may make it easier to follow (although this may change with editing).

Ans: We understand the issue but formatted the manuscript according to the guidelines for author. Most likely the article in its final format will be nicer to read.

27) In table 1a, the virulent column doesn’t really say much as many of the results are negative. Is this column worthwhile? Happy to be led by you, if you feel that you want it to stay then I wont argue with you.

Ans: Indeed, we think that this column is relevant because it clearly shows that only one animal (adult male Alpine ibex) was found to be positive for the virulent strain and showed severe lesions. It is striking when comparing this Table 1b, which presents the corresponding data in domestic animals. Since the upcoming nationwide control program in sheep will be only focusing on the virulent strain, table 1a clearly illustrates that wild ruminants should not be considered as a significant infection source for sheep and are therefore not expected to jeopardize the management measures that will be put in place for the Swiss sheep population.

28) The legends for table 1a and 1b would benefit from a bit more detail as to what the tables show. Also please define what SAC means in table 1b.

Ans: Done.

29) Line 241. A comma between S1 table) and all may aid flow here

Ans: Done.

30) Line 249, again, a comma between red deer and positive would aid flow.

Ans: Done.

31) Line 256, benign strain presented with severe

Ans: Done.

32) Line 264. Front feet of an ibex which was/ were positive. Alternatively a comma would suffice

Ans: Done.

33) And line 266- hind feet of an ibex which were positive. Alternatively a comma would suffice

Ans: Done.

34) Line 269- You discuss interspecies interactions. How clear would this be? Is it obvious and noticeable when they interact, or is it more that the animals were found in the same area? I think it is very interesting, but just want to try and understand it a little more.

Ans: Four different types of interactions were recorded according to the definitions below, relying on former observations of experienced field partners: 

1) Direct contact (i.e., physical contact). 

2) Indirect contact < and > 50 m of distance, respectively

3) Non simultaneous occupation of the same pasture

35) Line 282 and 283. You mention small domestic ruminants. Do you mean cattle and sheep here? Or others too? Could you be more specific?

Ans: Small domestic ruminants typically refers to sheep and goats. For more clarity, we replaced small domestic ruminants by "sheep and goats".

36) Line 283-284- As concerns seems slightly strange wording. Maybe with reference to?

Ans: Done.

37) Line 288. You switch her to domestic ungulates? Could you stick to the same terms as used previously, and maybe specific sheep and cattle, or others. Or even a link to table 1b may be useful?

Ans: We have taken into account your remark and changed the manuscript accordingly. We added specific clarification as to which wild and domestic ungulates we intended. 

Line 292-294:"Data on reported proximity among wild and domestic ungulates (ibex, chamois, red deer, roe deer, sheep, cattle, goats and South American camelids) are summarized in figure 4."

We however do not feel that a link to table 1b is useful here. 

38) Line 293- physical contacts. How do you define this?

Ans: The type of interspecies contacts that were used in this study rely on previous studies (field observations, questionnaire survey) conducted in the Canton of Grisons in Switzerland. Physical contacts mean direct body contact between two individuals (without any space between them).

39) Line 299. A pair of commas around except domestic goats, maybe help flow here

Ans: Done.

40) Line 317- what made you choose a cut off value of 0.2?

Ans: For our statistical analysis we decided to use a two-step approach. In a first step, we conducted a univariable analysis with a cut off value of 0.2. This value was chosen arbitrary, but lays within the range of commonly used values ranging from 0.10 to 0.25 (Bursace et. al 2008; Purposeful selection of variables in logistic regression, Source code for biology and Medicine; 3:17). The procedure consists in using a cut off that is less strict than 0.05 to be able to identify variables that are meaningful to further explore them. 

41) Table 2. If you could merge the top two lines where possible it would make the table clearer

Also it is clearer and more helpful not to leave gaps in the table, maybe an N/A would be more beneficial?

Ans: Done.

42) Line 330, you mentioned the species red deer- is that being of that species or encountering it?

Ans: The results of the multivariable analysis showed that red deer (as a species and not the encounter with red deer) were more likely to be carriers of D. nodosus in comparison to ibex.

43) Table 3. Maybe discuss in the legend what you mean by baseline?

Ans: Baseline is a commonly used term in risk factor analysis and signifies the level of comparison within an explored variable. We added in the table captions "(level of comparison)" to clarify this. 

44) Line 256. A full stop is needed between livestock species, and The comparison

Ans: Done.

45) Line 361- Insertion of strain after the word benign may aid reading

Ans: Done.

46) Line 372- ‘As for the other two Alpine ibex that harboured the benign strain’ This sounds a bit colloquial. Maybe consider rewording.

We reworded the sentence as follows: 

Line 389-390 " As for the other two Alpine ibex that were found positive for benign strains…"

47) Lines 382-389- you discuss the potential involvement of Treponema species. The academic editor handling this manuscript is an expert in Treponema species, so it maybe worth a discussion with him to see if he would be willing to run the samples for you as a collaboration to further this work? Just a thought

Ans: We have already considered this option and frozen samples of the available affected feet for potential later testing, but it is out of the scope of this manuscript. In a previous study conducted at our group on footrot in Alpine ibex (Wimmershof et. Al 2015; Outbreak of severe foot rot associated with benign Dichelobacter nodosus in an Alpine ibex colony in the Swiss Prealps, Schweiz Arch Tierheilkd; 157(5)) histology was performed on affected feet and there was no suspicion of Treponema sp. Nevertheless, we agree that PCR testing would be better, therefore the decision to freeze new samples for later testing.

48) Line 400- maybe be biased? But how?

Ans: Since the information collected on interspecies contacts relied mainly on memorized events of the professional participants it is possible that it may have resulted in misreporting or a lack of reporting. We rephrased the sentence for more clarity.

49) Line 400. You discuss the interactions, but would the presence of people (possibly hunters) affect this?

Ans: Since observations of interspecies interactions were not limited to the hunting season but reflect interactions throughout the year (except with domestic ungulates, present in wildlife habitat only during the grazing season), it is unlikely that hunting influenced the data. However, presence of people (tourists, observer) may of course have had an influence on interactions observed at other times of the year. Human disturbance may have prevented interactions rather than fostered them. Therefore, the reported occurrence of interactions is most likely underestimated; this is also true as the observers have not always been present in the field, and interactions may occur at night.

50) Line 404. You talk about interactions mainly being with cattle and sheep- but is this not due to the fact that these are the most abundant animals?

Ans: It is indeed true that the most abundant domestic ungulates in Switzerland are cattle and sheep. This however does not change the fact that most interactions were observed with these two species, meaning that pathogen transmission between wild ungulates and livestock is most likely to occur with sheep and cattle, i.e., with the ruminants susceptible to act as a source of D. nodosus infection.

51) Line 412-415- I read this a few times, but never fully understood what you were trying to say. Maybe consider rewording it?

Ans: We revised the sentence and added a punctuation for better word flow. 

52) Line 419- comma after nodosus would aid reading flow.

Ans: Done.

53) Line 434- avoid the use of our.

Ans: Done.

54) Line 457- does not necessitate to involve wildlife doesn’t seem to make sense, and is unclear- consider rewording

Ans: We have reworded this sentence as follows: 

Line 472 " … does not need specific wildlife management measures"

55) Line 458- you use prevalence previously to be plural, so I would avoid prevalence’s

Ans: Done.

56) Line 461- what do you mean by a sanitised herd?

We mean herds that have been already treated for footrot and have been tested negative for D. nodosus.

57) Figure 1 and 2. Would it be possible to combine these? Possibly by using a thicker black border around the circles which appear on both figures?

Ans: We would prefer to keep the images separate as it clearly demonstrates how few D. nodosus positive animals were found on the whole Swiss territory compared to the large amount and wide distribution of collected samples. Due to the marked overlapping of the collected samples (figure 1), the difference in numbers between D. nodosus positive and negative would not stand out as clearly if the two figures were merged. 

58) The graphs are a little unclear- but that may just be my print out

Ans: We re-checked the graph and noticed that Figure 4 was indeed of low resolution. We made the necessary changes and controlled again all figures with the PlosONE Figure check. 

Reviewer #2: 

A manuscript entitled “Identifying maintenance hosts and risk factors for infection with Dichelobacter nodosus in free-ranging wild ruminants in Switzerland: a prevalence study” had the goal of identifying cases of benign and virulent D. nodosus in common wild ungulates.

Strengths: The authors report the PCR results of 1821 sampled wild ungulates collected between 2017-2018, which is an impressive number of animals representing the most common wild ungulates with ranges covering most of Switzerland, and in a short enough time period, such that results likely reflect the general overall trend of this organism. The number of species sampled were mostly balanced. The method of foot swabbing, field preservation and D. nodosus PCR has been well elaborated in previous publications and was likely effective. Authors were able to deduce that prevalence of D. nodosus was very low in wild ungulates despite common wild -domestic ungulate interface in most of the regions sampled. This is novel and valuable information and timely as Switzerland moves to report and control virulent D. nodosus in domestic livestock. This work was done in parallel with similar published work on domestic livestock, which strongly demonstrates that domestic livestock prevalence of D. nodosus is common and significant in comparison to wild ungulates.

Weaknesses:

1) The title suggests that “risk factors” for Footrot in non-domestic ungulates will be evaluated, though risk factors are only loosely suggested in the text with no data in the study design to make an observation or conclusion as to risk. If this is meant to broadly encompass domestic-wild population interface, this should be worded differently.

Ans: Indeed our study design was not set up to evaluate risk factors for foot rot, but was calculated to estimate the prevalence of infection with D. nodosus in wild ungulates in Switzerland. Our focus was not disease but infection with D. nodosus. We attempted to identify risk factors for infection in a second step, although for wild ungulates it was only possible for begnin D. nodosus. We now changed the title of our study to: "Identifying maintenance hosts for infection with Dichelobacter nodosus in free-ranging wild ruminants in Switzerland: a prevalence study". We agree that it is better suited, since the "risk factors" part was only a supplementary analysis and not the main objective of our study. 

2) This manuscript relies heavily on observations and sample collection using untrained participants. This is worrisome for accuracy of specimens (were negative swabs truly negative cases?) and overall identification and interpretation of lesions, especially early lesions. This is particularly of concern when detailing wild – domestic animal interface, which was determined entirely from historic memory of untrained participants. The weaknesses of this data are well detailed by the authors in a paragraph starting Ln 395, and the authors should be careful to draw conclusions about transmission between livestock and wildlife, Ln 446. Further molecular work may need to be done comparing ibex and cattle / sheep isolates to make these conclusions.

Ans: In previous studies carried out by the FIWI, Swiss game wardens and hunters were asked to collect swab samples (conjunctiva, lung) for the detection of Mycoplasma conjunctivae and M. pneumoniae, revealing a high prevalence for both bacteria. The fact that the professionally trained game wardens (present in all large Swiss cantons) and hunters had already successfully contributed to similar studies, together with their basic education in wildlife health, makes us feel confident that they were capable to carry out the sampling correctly. 

Importantly, a detailed sampling instruction sheet was provided with every sampling kit, explaining how to take the samples, with detailed illustrations of the different steps.

Last but not least, to make sure that the samples would be collected following the same standards as in the parallel study conducted in domestic ungulates (Ardüser et. Al 2019; Dichelobacter nodosus in sheep, cattle, goats and South American camelids in Switzerland—Assessing prevalence in potential hosts in order to design targeted disease control measures; Prev Vet Med. ;104688) we gave lectures and courses in three of the largest Swiss cantons (Bern, Grisons and Vaud) before the start of the sampling season in 2017. In total, these three cantons made up 42 % of the collected samples. 

Actually, we suspect that prevalence is less likely to be underestimated in wildlife than in domestic animals, because wildlife feet (incl. the interdigital space) are generally clean, in contrast to domestic animal feet, which are often covered by mud and feces and need to be roughly cleaned before sampling. 

Regarding the identification and interpretation of the lesions, in every sampling kit a data sheet was provided where lesions of the interdigital space, the hooves and the carpal joints could be noted. It was presented as a checklist with the description of the different disease stages and associated lesions. This procedure has already been successfully used for another disease. It would have not been practicable to require from our field partners to submit every foot to be sampled at the lab and this additional work would have dramatically increased the risk that we would not achieve the targeted sample size. We therefore only asked of our field partners to send us feet of animals with foot lesions.

Overall, although we cannot guarantee that all the sampling procedures were always respected and that the detailed information was 100% correct, we are confident that the obtained data are reliable.

Regarding the observation of intraspecies interactions under free-ranging conditions, the professional training and field experience of game wardens is best suited and certainly adequate for the observation of wildlife. Here they are even expected to make more reliable observations than scientists, who are typically less familiar with the study area and less used to observe wildlife. We agree that reports based on memorized events always entail a risk of error. However, previous similar studies carried out by our group on interspecific interactions between Caprinae and Ovinae (Ryser-Degiorgis et. al 2002; Encounters between Alpine ibex, Alpine chamois and domestic sheep in the Swiss Alps; Hystrix the Italian Journal of Mammalogy; Vol 13) as well as well a study conducted in the Canton of Grisons regarding interactions at the wildlife livestock interface (Casaubon et. al 2012, Bovine viral diarrhea virus in free-ranging wild ruminants in Switzerland: low prevalence of infection despite regular interactions with domestic livestock; BMC Vet Res; 8:204) have provided former training in this field. Most recently, we could also show in a study on sarcoptic mange (Pisano et al. 2019, revised; Spatiotemporal spread of sarcoptic mange in the red fox in Switzerland over more than 60 years: lessons learnt from comparative analysis of multiple surveillance tools revised) that data retrospectively collected by questionnaire among field partners (memorized events) were reliable, by “validating” the data with another, independent information source.

The results we obtained on interspecies interactions are in agreement with previously conducted studies, which support the validity of our data.

We fully agree that further molecular work needs to be done to draw conclusions about the transmission of D. nodosus between livestock and wildlife. We therefore changed the text in the manuscript accordingly:

Line 459-462 " Therefore, cattle may act as source of infection for susceptible wild ungulates, as previously suspected (4). Nevertheless, further molecular studies are needed to univocally prove that interspecies transmission of D. nodosus occurs between wildlife and domestic livestock."

3) Authors attempt to correlate gross and lack of gross lesions with PCR results in wild ungulates but rely mostly on participants reports and a very small subset of feet submitted for professional evaluation. This limitation should be considered throughout the text when discussing the relevance of species contact.

Ans: We would like to point out that the majority of our samples (85% N = 1556) were collected by professionally trained game wardens who routinely help with the surveillance of wildlife diseases in Switzerland. See also the comment above (nr. 2). Out of 1821 samples, 76 animals were reported to have foot changes, consisting mainly in overgrown hooves. Out of these 76, five were positive for benign D. nodosus and one for virulent D. nodosus.

Severe footrot-like lesions were confirmed in two ibex via veterinary examination; as indicated in the manuscript, while marked lesions in two red deer could not be confirmed since these feet were unfortunately not sent for examination. 

While we agree that the data on foot lesions are few, we don’t understand in which regard the evaluation of foot lesions could affect the discussion on the relevance of species contacts. 

Line 395 – 397 " Two red deer with reported foot lesions tested positive for benign D. nodosus. Whether these lesions were footrot-like could not be confirmed because neither the affected feet nor photographs were submitted for veterinary evaluation."

 4) Discussion of pathogenesis and progression of foot lesions is not appropriate from the small subset of feet attained and is also not within the stated scope of the manuscript. The gross ibex foot lesions demonstrated are impressive and similar to that described by Wimmershoff et al 2015, but are these caused by D. nodosus? Histopathology, a larger subset of hooves representing various degrees of disease, along with a more detailed analysis of associated microorganisms would be necessary to call these primary D. nodosus lesions. D. nodosus associated? Please keep this deficit of information in mind when discussing disease development.

Ans: We agree with the fact that we cannot discuss the pathogenesis and progression of foot lesions from the small subset of samples we have obtained and adapted the passage as follows: 

Line 389-391 " As for the other two Alpine ibex that were found positive for benign strains without reported foot lesions but were not sent for veterinary examination, it cannot be ruled out that they may have been in an early stage of infection."

In the paper by Wimmershoff et al 2015, it was purposely not stated that the gross lesions observed in the two Alpine ibex were*caused* by D. nodosus but as you correctly pointed out, that they were *associated* with a D. nodosus infection. Furthermore, the potential involvement of other infectious agents is discussed.

Line 380 – 382 "These findings are in line with previous reports documenting that the presence of benign and virulent strains of D. nodosus can be*associated* with severe foot lesions in Alpine ibex that are comparable to typical lesions of footrot in sheep (1,4)."

We agree that we would need to have a greater subset of feet representing various degrees of disease severity and a more detailed analysis of associated microorganisms. Unfortunately, a larger subset of feet was not available to us since only a few animals with foot lesions were observed. From the four Alpine ibex found positive (1 virulent , 3 benign) only the limbs of the animals with severe lesions were sent to our laboratory for veterinary examination (2/4). It is important to remember that the Alpine ibex is a protected species and animals are culled only in case of severe disease signs. Since additional data collected within a side study indicate that footrot occurs only sporadically in ibex, it is unlikely that it will ever be possible to perform such a study, unless samples are stored over many years. In any case, a definitive proof of the causal relationship between D. nodosus (whether benign or virulent) and lesions on ibex feet could only be provided through an experimental infection. As regarding further analysis to rule out co-infection with other microorganisms, this would indeed be interesting but it is beyond the scope of our study, which aimed to determine the occurrence of D. nodosus in the Swiss wild ungulate population in order to better plan the upcoming nationwide control program in sheep. 

5) There is no mention of season of data collection and particularly seasonal correlation with positive D. nodosus cases in wildlife, though this would likely have an influence in disease occurrence and progression. Could this be determined from the data?

Ans: Please note that we mentioned the season of data collection in the Materials & Methods section: 

Line 118 – 119: "Sampling was carried out from August 2017 to December 2018, with most samples collected during the hunting seasons (i.e. August-December) of 2017 and 2018."

Regarding the seasonality, it would have indeed been interesting to examine whether D. nodosus carriage was associated with any season. However, on the one hand only very few samples turned out to be positive, making questionable an analysis for this variable. On the other hand, most of the samples (89 % N = 1622 ) were collected during the hunting season (August-December 2017-2018), preventing a meaningful grouping of samples according to seasons; finally, most (86 % N = 31/36) of the D. nodosus positive samples were also found during the hunting season. Overall, this variable could not be evaluated due to both the unbalanced number and the distribution of positive and negative samples. We tried to better explain the distribution of the collected samples over the year in the Materials and Methods section.

6) This study was done in parallel with a similar published study in domestic livestock. Where there is spacial overlap of wild D. nodosus cases with positive domestic cases, would it be possible to perform more detailed molecular work (sequencing, etc.) comparing the wild and domestic D. nodosus to demonstrate or exclude the possibility of a shared microorganism passing from domestic animals to wildlife?

Ans: Given the sampling plan (samples from all over the country in both wild and domestic ungulates) and the data collected on observed interactions, we consider indeed that there was a spatial overlap. As concerns potential molecular work, please see the end of the comment above (nr. 2).

Below are addition comments:

7) Ln 16 Footrot is a disease of feet and not specifically the hoof. Unless the lesion is being described only within the structures of the hoof, these references should be changed to foot / feet.

Ans: Done.

8) Ln 18, 51, 256 Please add …presenting “with” mild and …..

Ans: Done.

9) Ln 19 The sentence starting “In Switzerland..” seems out of place, as no correlation is established in the abstract between Ibex and domestic sheep. The abstract needs to be organized so that this information falls into logical place and leads to why wild ruminants are to be examined.

Ans: We would like to keep this sentence because it is the link between the wild and domestic investigations and it shows the main motivation behind the study (future upcoming nationwide footrot control program in sheep). The main objective was therefore to investigate whether other ungulates (domestic and wild) possibly susceptible to D. nodosus could jeopardize the upcoming control program. 

We have however adapted the sentence to try to make the transition smoother: 

Line 18-20: "Because the disease is widespread throughout sheep flocks in Switzerland, a nationwide footrot control program for sheep focusing on virulent strains shall soon be implemented."

10) Ln 21 Please clearly note if “maintenance hosts” are wild or domestic or both, and if “sheep” are domestic or wild.

We modified the manuscript accordingly: 

Line 22 – 23 " …potential susceptible wildlife maintenance hosts that could be a reinfection source for domestic sheep."

11) Ln 31 This appears a big jump to implement domestic animals as the cause of disease in wild animals. That was not the focus of the study and appears to be something that would need to be proven under controlled lab conditions and with specific molecular work to prove identical type stains. Noting that there is a low incidence of wild carriers and results in severe disease in wild hoofstock appears more in line with the aims and outline work, at least in the abstract.

Ans: 

We modified the sentence in the abstract as follows:

Line 30-31 " In conclusion, the data suggest that wild ungulates are likely irrelevant for the maintenance and spread of D. nodosus.

We discuss the potential transmission from domestic to wild ruminants in the discussion because it is very relevant to disease management in ibex, which is a protected species. So far, sheep have been incriminated as the source of infection and managers considered separating more strictly ibex from sheep, but our study reveals that as concerns the benign strain group, cattle are more likely to be the source of infection. We fully agree that a proof for this suspected scenario can't be delivered till further molecular work is done. Please note that we don’t speak of proof in our manuscript but of suggestions and likelihood

See also the end of comment nr. 2 as concerns molecular work.

12) Ln 45 Please change to “practical”

Ans: Thank you for this suggestion. However, we have decided to change the word "practicable" into "feasible" which we think is more appropriate in its meaning. 

13) Ln 51 Introduce the molecular or other difference between “benign” and “virulent” strains here. How do you differentiate these organisms.

Ans: We modified the sentence as follows: 

Line 48-52 "In sheep, mild and severe clinical forms of footrot have been shown to be associated with the presence of strains of D. nodosus carrying either the AprB2 or AprV2 genes, respectively, which encode for the subtilisin-like extracellular proteases playing a key role as virulence factors (8,10), thus referred to as "benign" and "virulent". We also added additional information about the definition of mild and severe lesions.

14) Ln 56 How does co-infection play into D. nodosus infection and susceptibility in domestic animals? This would be appropriate to discuss here.

Ans: We have added a sentence as suggested. 

Line 59-61 " In sheep, co-infection with anaerobic bacteria such as Fusobacterium necrophorum has been suggested to increase disease severity (9,11,16)."

15) Ln 72 Replace “of them” with “hoof stock”

Ans: Done.

16) Ln 131 Do the authors have a sense of reproducibility and agreement on lesion scoring, description and swabbing technique between game wardens, hunters, etc.? If so, please note how that agreement was determined. My experience is that there is a very big difference between lesion grading done by biologists, lay people and veterinarians and particularly with interpretation of early lesions. Were wardens and hunters trained in appropriate swabbing and handling techniques? If not, could this have confounded the findings? Explain.

Ans: We did not calculate an agreement among the results obtained by sample collectors with different professional backgrounds for this study but as stated in comment nr. 2 we have good reasons to believe that the data are reliable. 

17) Ln 134 Was season taken into account for sampled collected? Was this information recorded? It seems environmental conditions would play importantly into risk of foot rot and general overall health of the animal.

Ans: Yes, the date of sample collection was systematically recorded. See also comment nr. 5. For more clarity, we slightly modified the text as follows:. 

Line 136- 137 : " …a data sheet to record the sampling date, biological data of the animal sampled (species, sex, age, geographical origin, body condition) and the presence of foot lesions".

Winter conditions typically impact the general health status of wild ungulates (esp. body condition due to poor food resources), which could result in higher susceptibility to infection; spring and fall are expected to be more humid, i.e. more favorable to D. nodosus survival; summer is the period when wild ungulates co-graze with domestic animals, i.e. when the exposure to D. nodosus is expected to be highest, but most PCR-positive and the two diseased ibex ( both adult males) were found during the hunting season, i.e. right after the period with the highest exposure to domestic ruminants. Anyway, we could not carry out a risk analysis for footrot with just two cases of disease, and it would not even be possible for PCR-positive animals.

18) Ln 138 How were affected feet submitted? What was time elapsed between collection and receipt in the lab and how were the feet handed / analyzed once in the lab?

Ans: The affected feet were submitted via rapid post mail (delivery within 2 days) right after collection. Upon receival we examined the feet (veterinary macroscopic examination), filled out the animal data sheet like our partners in the field, took the swabs (as specified in the protocol that was also sent to the sample submitters). We then took photographs of the lesions and stored samples in a -20° C freezer as well as in formalin (for potential later additional investigations, see comment nr. 4). The swabs were then transmitted directly to the Institute for Veterinary Bacteriology (IVB) at the Vetsuisse faculty of the University of Bern (located just a floor above our institute) for DNA extraction and subsequent Real Time PCR analysis.

The average time elapsed between swab collection in the field whose feet were not sent to the lab (i.e., mostly in animals without lesions) and shipment to the lab was of 4.53 days. The samples (lysis buffer) were then directly stored (refrigerator) at the IVB and the DNA extracted. The time elapsed between arrival of the lysis buffer at the laboratory and Real Time PCR analysis was in average 67 days. 

19) Ln 139 Is this a historical report generated by the participant? Over what period of time are these observations of a 5 km radius representing? At the time of animal collection? Over what season?

Ans: The reports on interspecies interactions rely on memorized events (unlimited time period) of the professional game wardens. However, recent episodes are most likely reported, especially for young game wardens who have been in service for only a few years. The participants were asked to report interspecies interactions in the area where they had sampled the animal regarding the sampled species. 

We did not ask to specify during which seasons the observations needed to be made, as it was a simplified questionnaire, but interactions with domestic ruminants are expected to occur mostly (if not only) during the summer grazing season (June till September).

20) Ln 229, 237 These table legends should stand alone and clearly delineate these as Swiss wildlife and domestic ungulates, and the period for which the data was collected. Are these domestic ungulate results taken in the same time interval as wildlife? This will be important to state or explain how different time points are comparable.

Ans: We changed the table legend as suggested. 

Wild ungulates were sampled during two consecutive hunting seasons (August till December of 2017 and 2018). Further few samples were also collected throughout the year to make sure the sample size was achieved. 

Samples from domestic ungulates were collected in parallel to ensure the comparability of the results but for logistical reasons (large number of farms to be visited; animals always accessible) sampling took place all year round (from May 2017 till June 2018). Therefore, all seasons could be taken into consideration in the domestic animal part of the study. 

We added clarification to this sentence in the discussion: 

Line 372-374 " Importantly, the study was conducted in parallel to a nationwide survey on infections with D. nodosus in possibly susceptible domestic livestock species (May until June 2017- 2018) (14)."

21) Ln 256 Were gross feet available to the authors in all cases with positive D. nodosus PCR? Or is this statement based on participant reports?

Ans: We asked participants to submit feet only in case of gross lesions (see comment nr. 2) . Therefore, from the 37 PCR-positive wild animals (36 benign and 1 virulent) we received the feet from 3 animals only: the single PCR-positive for the virulent strain (ibex with severe foot lesions) and two PCR-positive for the benign strain, one of which concerned an ibex with severe lesions and the other a red deer with changes concerning the lateral digits only ). 

To make this clearer, we modified the sentence accordingly. 

Line 270-273 " Two of them were reported to present with changes consisting in overgrown hooves, one of which was submitted for veterinary examination. The anomaly concerned the lateral digits only, without noticeable interdigital inflammation (i.e. no signs suggestive of footrot)."

22) Ln 263 Were cases only evaluated by PCR for D. nodosus with gross examination of a small subset of cases with lesions?

Ans: We analyzed all samples that were submitted to us (n= 1821 swabs), most of which originated from wild animals without noticeable lesions. Of all sampled animals, 3.3% (N = 60) were sampled by veterinarians (FIWI), including a thorough inspection of the feet. However, all field participants were asked to follow the same detailed sampling protocol, data sheet and checklist to assess the presence of feet lesions See also comment nr 2 for more details on data collection. 

23) Ln 264 These are impressive gross lesions. For the outlined study, finding of D. nodosus and specifically the virulent strain is important, but what other organisms are associated with this lesion? Treponema? Fusobacterium? Most foot diseases are far more complex than a single microorganism. Perhaps beyond the scope of this article, but it would be nice to see representative sections of this lesion histologically and perhaps demonstrating the microorganism and any associated pathogens. What time of year was this animal collected? Were there other conditions found on processing the cadavers?

Ans: It is true that in foot diseases multiple organisms often play a role, such as Fusobacterium or Treponema. We try to underline this fact in the discussion, which we have completed as follows: 

Line 401-407 "In cattle and free-ranging reindeer, foot diseases with similar clinical findings can involve pathogens such as Fusobacterium necrophorum (cattle and reindeer) and Treponema sp. (cattle) (37–39). … "Therefore, since all samples of the present study were tested via qPCR for the presence of D. nodosus only, we cannot exclude that other pathogens causing foot disease in domestic and wild ruminants may have been involved in the development of the observed lesions."

Regarding the histology, Wimmershof et al. presented the microscopic lesions found in ibex with severe lesions associated with benign D. nodosus. No other infectious agents were detected in the tissue sections. See also comment nr. 4.

As for the adult male ibex with severe lesions and PCR-positive for virulent strains, the animal was sampled in late summer 2017 (23.08.2017), and the adult male ibex with severe lesions PCR-positive for benign strains in the autumn of 2018 (20.10.2018). 

Ibex are large animals living at high altitude, with males bearing large horns and weighting up to 80-100 kg. For logistical reasons we did not ask for the whole carcass and could not perform a complete pathological examination of the affected animals. 

The submitters of the samples from the two ibex positive to D. nodosus (virulent and benign strains) both mentioned a moderate body condition and severe feet lesions but no other abnormalities. 

24) Ln 269 The observations by participants is helpful, but could there not be structured information by trained wardens / biologists to more objectively report wild and domestic animal interface in these cantons? Clarify here as well, during what time period and season are these observations taken? Make sure that abbreviations are decoded in a foot note on all figures and tables and that titles and legends are descriptive sufficiently so that the items can stand alone when read separate from the text.

Ans: Most observations reported by our field partners were from professionally trained game wardens (85 % ;N = 1556), with a few number from wildlife biologists (4.5% ;N = 83). This is indicated in the Materials and Methods section.

Line 165-167. " In total 1,821 samples were taken, of which 91.4% (N = 1,664) by game wardens/hunters; 4.5% (N = 83) by field biologists; 3.3% (N = 60) by veterinarians (FIWI) and for 0.7% (N = 14) of the samples the profession of the sampler was not given."

There were former field studies performed by people of our team as well as by other scientists (Ryser-Degiorgis et. al 2002, Encounters between Alpine ibex, Alpine chamois and domestic sheep in the Swiss Alps, Hystrix the Italian Journal of Mammalogy; Vol 13, Richomme et. al 2006, Contact rates and exposure to inter-species disease transmission in mountain ungulates, Epidemiol. Infect., 134; R. Tschopp et. al 2005, Outbreaks of infectious keratoconjunctivitis in alpine chamois and ibex in Switzerland between 2001 and 2003, Veterinary Record, Vol 157); which were congruent with data recorded among game wardens by questionnaire, in this and in former studies (e.g. Casaubon et. al 2012, Bovine viral diarrhea virus in free-ranging wild ruminants in Switzerland: low prevalence of infection despite regular interactions with domestic livestock; BMC Vet Res; 8:204refs). For the present study, which was carried out in all of Switzerland, it would have not been doable to have scientists in all sampling areas to record occasional interspecies interactions. Based on own experiences, a questionnaire survey among wildlife professionals is the most efficient way to gather sufficient information on the occurrence of such interactions (Ryser-Degiorgis et. al 2002, Encounters between Alpine ibex, Alpine chamois and domestic sheep in the Swiss Alps, Hystrix the Italian Journal of Mammoogy, Vol 13; Casaubon et. al 2012, Bovine viral diarrhea virus in free-ranging wild ruminants in Switzerland: low prevalence of infection despite regular interactions with domestic livestock, BMC Vet Res 8:204).

For the questions on the sampling, please see comment nr. 5. Whenever appropriate, we adapted figure titles and legends as requested. 

25) Ln 386 This citation should include Treponema associated hoof disease in free-ranging elk Vet Pathol 2019.

Ans: Done.

26) Ln 390 Since this work evaluated a one-time PCR for D. nodosus with no controlled gross observations and missing data on subjects, it does not seem appropriate to discuss pathogenesis and progression of D. nodosus lesions in ibex.

Ans: We agree that from the few samples we have obtained we cannot discuss pathogenesis and progression of D. nodosus lesions in ibex. It had not been our objective but in accordance with your comment, we removed the following paragraph (Line 438-441; track change manuscript). 

" The presence of foot lesions was identified as a factor associated with the infection of benign D. nodosus. Obviously, the reported lesions were rather the consequence of the infection than a risk factor. This result from the risk factor analysis is also questionable because these lesions were only confirmed as footrot-like in the ibex."

We also decided to remove the lesions as a variable from our risk analysis since reported lesions were not consistent with footrot lesions or, if reported as consistent, could not all be confirmed by veterinary examination.

---

## [Decision Letter · Decision Letter 1]

12 Nov 2019

PONE-D-19-18147R1

Identifying maintenance hosts for infection with Dichelobacter nodosus in free-ranging wild ruminants in Switzerland: a prevalence study

PLOS ONE

Dear Marie-Pierre Ryser-Degiorgis

Thank you for submitting your manuscript to PLOS ONE. After careful consideration, we feel that it has merit but does not fully meet PLOS ONE’s publication criteria as it currently stands. Therefore, we invite you to submit a revised version of the manuscript that addresses the points raised during the review process. Please note the revisions are very minor.

Many thanks for your resubmission to PLOS One

The manuscript was reviewed by the same reviewers as the original, and one came back with some very minor comments- mainly typographical and grammatical.

If you could modify these minor points and resubmit it, I can then recommend it for publication.

I would do them for you, but I cant edit a PDF.

Please don't worry about a response to reviewers letter. If you can just write that all comments were addressed, then I can have a quick read and expedite its publication

Many thanks

Simon

We would appreciate receiving your revised manuscript by Dec 27 2019 11:59PM. To enhance the reproducibility of your results, we recommend that if applicable you deposit your laboratory protocols in protocols.io, where a protocol can be assigned its own identifier (DOI) such that it can be cited independently in the future. For instructions see: http://journals.plos.org/plosone/s/submission-guidelines#loc-laboratory-protocols

A marked-up copy of your manuscript that highlights changes made to the original version. This file should be uploaded as separate file and labeled 'Revised Manuscript with Track Changes'.An unmarked version of your revised paper without tracked changes. This file should be uploaded as separate file and labeled 'Manuscript'.

We look forward to receiving your revised manuscript.

Kind regards,

Simon Russell Clegg, PhD

Academic Editor

PLOS ONE

Reviewers' comments:

Reviewer's Responses to Questions

**Comments to the Author**

1. If the authors have adequately addressed your comments raised in a previous round of review and you feel that this manuscript is now acceptable for publication, you may indicate that here to bypass the “Comments to the Author” section, enter your conflict of interest statement in the “Confidential to Editor” section, and submit your "Accept" recommendation.

Reviewer #1: All comments have been addressed

Reviewer #2: All comments have been addressed

2. Is the manuscript technically sound, and do the data support the conclusions?

Reviewer #1: Yes

Reviewer #2: Yes

3. Has the statistical analysis been performed appropriately and rigorously? 

Reviewer #1: Yes

Reviewer #2: Yes

4. Have the authors made all data underlying the findings in their manuscript fully available?

Reviewer #1: Yes

Reviewer #2: Yes

5. Is the manuscript presented in an intelligible fashion and written in standard English?

Reviewer #1: Yes

Reviewer #2: Yes

6. Review Comments to the Author

Reviewer #1: Thank you for making the changes suggested last time. I really enjoyed reading the manuscript again. I have made a few more, very minor comments, but I don’t feel that I need to review it again. They are mainly just grammatical bits

Line 40 – concerning may be better worded as affecting

Line 50- the gene names should be in italics

Line 54- have been detected in the absence of ….

Line 117- on an animal level

Line 123- For all species, the design prevalence

Line 169-170- maybe reword this sentence as starting with a decimal place in words seems strange

Line 174- for roe deer and red deer, this was …

Line 181- a manufacturer and place of manufacture for the Kingfisher product maybe good

Laboratory analysis section- the reagents used for the PCR maybe useful

Line 206- 5 should be in words

Throughout, every time you use et al it should be followed by a full stop and a comma and in italics

Line 305- comma after goats (9%)

Line 351- risk factor

Table 3 looks untidy. Can it be tidied up so there isn’t a lot of empty boxes?

Line 411- dawn or dusk (xx, xx), it is…..

Line 431- smaller wildlife populations, it has been previously …

Line 449- remove allow to

Reviewer #2: This article “Identifying maintenance hosts for infection with Dichelobacter nodosus in free-ranging wild ruminants in Switzerland: a prevalence study” is a revision of a previously submitted manuscript with major revisions.

The manuscript has a clear and important objective to determine the prevalence of D. nodosus in Swiss wild ruminants, which it accomplished with an impressive number of animal samples covering much of the country, and concluding that wild ruminants in fact have a very low level of D. nodosus, compared to a sister study of domestic ruminants, which showed a high prevalence of the bacteria.

This manuscript is well written and used established protocols for collection and processing of samples. The interpretation of the data is legitimate and well-represented with statistically sound and significant conclusions “In conclusion, the data suggest that wild ungulates are likely irrelevant for the maintenance and spread of D. nodosus. Furthermore, we add evidence that both D. nodosus strain types can be associated with severe disease in Alpine ibex.”

The authors did a thorough job of addressing all reviewers’ comments with excellent elaboration of justifications and many appropriate clarifications and improvements to the manuscript. This is much appreciated and this manuscript will be an excellent addition to the current literature on foot disease in wild ruminants.

A minor point that may be detected by the photo editor, on Fig. 3 there is significant variation on white balance in the hoof background. Perhaps this can be tweaked to be better balanced? The lesions are well represented.

7. PLOS authors have the option to publish the peer review history of their article (what does this mean?). If published, this will include your full peer review and any attached files.

Reviewer #1: No

Reviewer #2: No

---

## [Editor Report · Decision Letter 2]

17 Dec 2019

Identifying maintenance hosts for infection with Dichelobacter nodosus in free-ranging wild ruminants in Switzerland: a prevalence study

PONE-D-19-18147R2

Dear Dr. Ryser-Degiorgis,

We are pleased to inform you that your manuscript has been judged scientifically suitable for publication and will be formally accepted for publication once it complies with all outstanding technical requirements.

With kind regards,

Simon Russell Clegg, PhD

Academic Editor

PLOS ONE

Additional Editor Comments (optional):

Many thanks for your resubmission to PLOS One

I have recommended your manuscript for publication. thank you for completing the comments which the reviewers stated last time.

I wish you the best of luck for your future research, and will keep an eye out for future papers from your group

Many thanks

Simon
---

## [Editor Report · Acceptance letter]

23 Dec 2019

PONE-D-19-18147R2 

Identifying maintenance hosts for infection with *Dichelobacter nodosus* in free-ranging wild ruminants in Switzerland: a prevalence study 

Dear Dr. Ryser-Degiorgis:

I am pleased to inform you that your manuscript has been deemed suitable for publication in PLOS ONE. Congratulations! Your manuscript is now with our production department. 

With kind regards,

on behalf of

Dr. Simon Russell Clegg 

Academic Editor

PLOS ONE